# A Permafrost Implementation in the Simple Carbon-Climate Model Hector v.2.3pf

Dawn L. Woodard[1], Alexey N. Shiklomanov[2], Ben Kravitz[3,4], Corinne Hartin[1], and Ben Bond-Lamberty[1]

[1]Joint Global Change Research Institute, Pacific Northwest National Laboratory, College Park, MD, USA 20740
[2]NASA Goddard Space Flight Center, Greenbelt, MD, USA 20771
[3]Department of Earth and Atmospheric Sciences, Indiana University, Bloomington, IN, USA 47405.
[4]Atmospheric Sciences and Global Change Division, Pacific Northwest National Laboratory, Richland, WA, USA 99352

**Correspondence:** Dawn L. Woodard (dawn.woodard@pnnl.gov)

**Abstract.** Permafrost currently stores more than a fourth of global soil carbon. A warming climate makes this carbon increasingly vulnerable to decomposition and release into the atmosphere in the form of greenhouse gases. The resulting climate feedback can be estimated using land surface models, but the high complexity and computational cost of these models make it challenging to use them for estimating uncertainty, exploring novel scenarios, and coupling with other models. We have added a representation of permafrost to the simple, open-source global carbon-climate model Hector, calibrated to be consistent with both historical data and 21st century Earth system model projections of permafrost thaw. We include permafrost as a separate land carbon pool that becomes available for decomposition into both methane ($CH_4$) and carbon dioxide ($CO_2$) once thawed; the thaw rate is controlled by region-specific air temperature increases from a pre-industrial baseline. We found that by 2100 thawed permafrost carbon emissions increased Hector's atmospheric $CO_2$ concentration by 5-7% and the atmospheric $CH_4$ concentration by 7-12%, depending on the future scenario, resulting in 0.2-0.25 °C of additional warming over the 21st century. The fraction of thawed permafrost carbon available for decomposition was the most significant parameter controlling the end-of-century temperature change in the model, explaining around 70% of the temperature variance, and distantly followed by the initial stock of permafrost carbon, which contributed to about 10% of the temperature variance. The addition of permafrost in Hector provides a basis for the exploration of a suite of science questions, as Hector can be cheaply run over a wide range of parameter values to explore uncertainty and easily coupled with integrated assessment and other human system models to explore the economic consequences of warming from this feedback.

## 1 Introduction

Permafrost—soil that continuously remains below 0°C for at least two consecutive years—underlies an area of 22 ($\pm$ 3) million km[2], roughly 17% of the Earth's exposed land surface (Gruber, 2012), and is estimated to contain 1460-1600 Pg of organic carbon (Schuur et al., 2018). Recent increases in global air temperature (Stocker et al., 2013), which are amplified at high latitudes (Pithan and Mauritsen, 2014; Biskaborn et al., 2019), have resulted in widespread permafrost thaw (Romanovsky et al., 2010), and simulations from variety of climate and land surface models across a wide range of scenarios suggest that this trend will continue into the future (Koven et al., 2013; Chadburn et al., 2017).

As permafrost thaws, its carbon becomes available to microbes for decomposition, resulting in the production of carbon dioxide ($CO_2$) and methane ($CH_4$) (Treat et al., 2014; Schädel et al., 2014; Schädel et al., 2016; Bond-Lamberty et al., 2016; Nzotungicimpaye and Zickfeld, 2017) that could lead to further warming (Koven et al., 2011; Schuur et al., 2015). Accounting for this permafrost carbon-climate feedback generally increases projections of greenhouse gas concentrations and global temperatures (Schuur et al., 2015; Burke et al., 2020) and increases estimates of the economic impact of climate change (Hope and Schaefer, 2015; Yumashev et al., 2019; Chen et al., 2019). However, the magnitude of this feedback is still highly uncertain, due to limited data availability and missing process-based understanding (Burke et al., 2017, 2020). The potential impact ranges from negligible to large, with stronger effects possible particularly over longer time horizons (Schuur et al., 2015).

Land surface models, like the Community Land Model (CLM) and the Joint UK Land Environment Simulator (JULES), use process-based representations of permafrost and explicitly model relevant components such as soil heat flux, soil moisture, hydrology, and vegetation and output thaw extent and depth, as well as emissions from permafrost soils (Chadburn et al., 2015; Lawrence et al., 2012). While high complexity models benefit from uncertainty quantification, they require large numbers of inputs and are computationally expensive, making it difficult to do uncertainty analysis directly with these models.

Conversely, simple climate models such as the Model for the Assessment of Greenhouse Gas Induced Climate Change (MAGICC) (Meinshausen et al., 2011) and Hector (Hartin et al., 2015) sacrifice spatiotemporal resolution and de-emphasize process realism in favor of conceptual simplicity and fast execution time. As a result they can be used to explore permafrost effects over a wide range of parameters and to analyze the relative significance of various permafrost controls. Similar models have previously been used to explore permafrost processes such as abrupt thaw that are not yet included in Earth system models (ESMs) (Turetsky et al., 2020) and to understand structural and parametric uncertainty (Schneider von Deimling et al., 2015; Chadburn et al., 2017; Koven et al., 2015b). Simple climate models can also be calibrated to emulate the mean global behavior of Earth system models to a high degree of accuracy (Meinshausen et al., 2011).

Here we describe the addition of a permafrost pool and a permafrost thaw mechanism to the simple carbon-climate model Hector, with the goal of providing a long-term platform for addressing a suite of science questions. Hector has been used for a wide range of analyses including climate effects on hydropower (Arango-Aramburo et al., 2019), ocean acidification (Hartin et al., 2016), global building energy use (Clarke et al., 2018), and for exploring the effects of observational constraints on estimates of climate sensitivity (Vega-Westhoff et al., 2019). Including a representation of permafrost in this model will allow for the consideration of permafrost in future such analyses with Hector, and, thanks to Hector's ability to represent separate biomes or regions, will be particularly important for evaluating the specific impacts of climate change in high latitudes.

## 2   Hector Model Design

Hector (Hartin et al., 2015, 2016) is an open source, object-oriented simple carbon-climate model that can emulate the global-scale behavior of more sophisticated climate models. Hector's simplicity and modular design make it easy to change the model's internal structure, while its fast computation time (~1-2 seconds) allows for easier interpretation of model behavior and facilitates sensitivity and uncertainty analyses, as well as prototyping of new submodules and features. Other significant

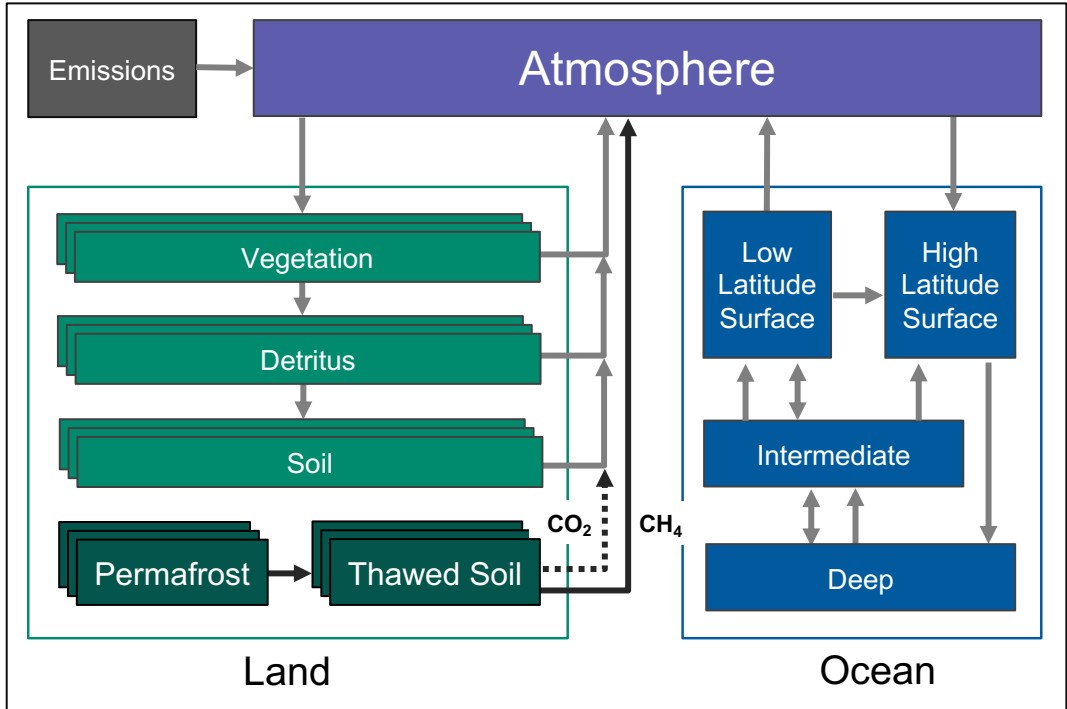

**Figure 1.** Hector's default carbon cycle showing fluxes (arrows) between each carbon pool. The terrestrial carbon cycle pools can be split into multiple regions, biomes, or other user-defined categories, so these are shown with multiple boxes. In darker green we show the addition of our novel permafrost representation in Hector. As carbon is exchanged in a variety of forms in Hector, the carbon flux arrows do not correspond to any particular carbon compound except where specified for land emissions. Vegetation, detritus, soil all emit $CO_2$, while thawed soil produces both $CO_2$ and $CH_4$ emissions.

advantages of Hector are its low memory requirements, ease of compilation, and optional R interface for setting inputs and parameters and retrieving model outputs. We focus here on Hector's carbon cycle as relevant to the addition of a permafrost carbon pool, but for a detailed description of the structure, components, and functionality of the base version of Hector see

Hartin et al. (2015). For subsequent updates, see the Hector GitHub repository (https://github.com/JGCRI/hector).

Ocean carbon in Hector is exchanged between the atmosphere and four carbon pools that model both physical circulation and chemical processes in the ocean. Carbon is taken up from the atmosphere in the high latitude surface box, which transfers some portion of this carbon to the deep ocean carbon pool. Carbon then circulates up to the intermediate ocean layer, to the high and low latitude surface pools, and is then outgased back to the atmosphere from the low latitude surface pool (Figure 1).

Hector's default terrestrial carbon cycle includes three land carbon pools: vegetation, detritus and soil, which can each be separated across multiple user-defined categories (corresponding to, e.g., biomes, latitude bands, or geopolitical units), each with their own set of parameters. When speaking generally, we will refer to these categories as 'groups' in this text. The vegetation pool takes up carbon from the atmosphere as net primary productivity (NPP), some of which is transferred into the

detritus pool, which can be decomposed and enter the soil carbon pool. All three land carbon pools separately emit carbon
back to the atmosphere from land use change, and soil and detritus release additional carbon through decomposition-driven
microbial respiration (Figure 1).

The annual change in atmospheric carbon in Hector, $\frac{dC_{atm}}{dt}$, at time $t$ in units of petagrams of carbon per year is given by:

$$\frac{\Delta C_{atm}}{dt}(t) = F_A(t) + F_{LC}(t) - F_O(t) - F_L(t) \tag{1}$$

where $F_A$ is the flux of anthropogenic industrial and fossil fuel emissions and $F_{LC}$ is land use change emissions, both
defined as positive to the atmosphere. $F_O$ is the net atmosphere-ocean carbon flux, and $F_L$ is the land-atmosphere carbon flux,
both defined as positive into their respective pools. $F_L$ is defined as NPP (carbon uptake) minus emissions from heterotrophic
respiration (RH) at time $t$ across all $n$ number of user-defined groups:

$$F_L(t) = \sum_{i=1}^{n} \text{NPP}_i(t) - \sum_{i=1}^{n} \text{RH}_i(t) \tag{2}$$

Heterotrophic respiration for group $i$ at time $t$ ($RH[i,t]$, $\frac{PgC}{yr}$) includes contributions from both soil ($RH_s$) and detritus
($RH_d$) decomposition, though only includes emissions from $CO_2$, not $CH_4$:

$$RH[i,t] = RH_s[i,t] + RH_d[i,t] \tag{3}$$

$$RH_d[i,t] = f_{rd} C_d Q_{10}[i]^{T[i,t]/10} \tag{4}$$

$$RH_s[i,t] = f_{rs} C_s Q_{10}[i]^{T_{200}[i,t]/10} \tag{5}$$

Detritus and soil heterotrophic respiration are both proportional to the sizes of their respective carbon pools ($C_d$ and $C_s$,
both in Pg C), with a rate that increases exponentially with temperature according to a group-specific temperature sensitivity
parameter ($Q_{10}[i]$). The corresponding fractions of respiration carbon, transferred annually, from each pool are given by $f_{rs}$
and $f_{rd}$. Detritus respiration increases with group-specific air temperature change ($T[i,t]$), while soil respiration increases with
the 200-year running mean of air temperature ($T_{200}[i,t]$), a somewhat arbitrary choice of smoothing used in Hector as a proxy
for soil temperatures in Hector's respiration calculations. This dampens the variability and produces a slower response in soil
warming compared to air temperatures.

$T[i,t]$ is the change in annual mean temperature (K) in group $i$ at time $t$ since the initial model period and is modeled as the
globally averaged mean annual temperature, $T$, at time $t$ multiplied by a group-specific warming factor, $wf_i$, that is set to 1 by
default for all groups but can be adjusted by the user:

$$T[i,t] = wf_i \cdot T[t] \tag{6}$$

## 2.1 Permafrost Submodel

We added permafrost to Hector as an additional, separate soil carbon pool that does not decompose or otherwise interact with
the rest of Hector's carbon cycle until it thaws. Hector's land carbon cycle with permafrost therefore includes five pools: vege-
tation, detritus, non-permafrost soil, permafrost, and thawed permafrost. Following previous modeling approaches, we focus on

only the top 3 m of permafrost (Kessler, 2017; Koven et al., 2015b), which is also consistent with the non-permafrost soil carbon pools in Hector. At each time step, a temperature-controlled fraction of permafrost carbon by mass is exchanged between the permafrost and thawed permafrost carbon pools. In the thawed permafrost pool, carbon is available for decomposition into $CO_2$ and $CH_4$ after subtracting a separately tracked stock of non-labile, or static, carbon in this pool. We define this static carbon fraction within the thawed permafrost pool following Schädel et al. (2014) as thawed permafrost carbon that is nearly inert and has a turnover time of up to thousands of years. Carbon moves primarily from the permafrost pool to the thawed pool as temperatures rise in the future, but refreeze of thawed carbon is also possible in scenarios where emissions reductions allow for potential cooling.

For a permafrost carbon pool at time $t$, $C_{perm}[t]$, and a thawed permafrost carbon pool, $C_{thawed}[t]$, (both in units of Pg C), permafrost carbon in Hector is exchanged as:

$$C_{perm}[t] = C_{perm}[t-1] - \Delta C_{perm}[t] \tag{7}$$

$$C_{thawed}[t] = C_{thawed}[t-1] + \Delta C_{perm}[t] - F_{thawed-atm} \tag{8}$$

where $\Delta C_{perm}[t]$ is the change in the permafrost carbon pool at time $t$ due to permafrost thaw or refreeze and $F_{thawed-atm}$ is the flux of carbon, in Pg C, from the thawed permafrost pool to the atmosphere, including both $CO_2$ and $CH_4$ emissions (see section 2.1.1). Assuming a uniform permafrost carbon density, $\Delta C_{perm}[t]$ is given by:

$$\Delta C_{perm}[t] = (f_{frozen}[t] - f_{frozen}[t-1]) \cdot C_{perm}[t-1] \tag{9}$$

where $f_{frozen}[t]$ is the mass fraction of permafrost carbon remaining at time $t$.

To a first approximation, $f_{frozen}[t]$ can be estimated as a function of mean air temperature (global or adjusted by a group-specific warming factor). We calculate $f_{frozen}$ at each time step in Hector following the model reported by Kessler (2017), but we recalibrated the model to use high latitude temperatures, $T_{HL}$ (which are proportional to global temperatures based on a high latitude warming factor, $wf_{HL}$), instead of global mean surface temperatures, and we use a lognormal cumulative distribution function (CDF) instead of a linear model.

$$f_{frozen}[t] = 1 - \text{NCDF}(\log(\Delta T_{HL})|\mu, \sigma) \tag{10}$$

$$T_{HL}[t] = wf_{HL} \cdot T[t] \tag{11}$$

where $NCDF$ is the normal cumulative distribution function and $\mu$ and $\sigma$ are the mean and standard deviation of the lognormal distribution. These two parameters control the frozen fraction of permafrost as a function of temperature and can be interpreted as follows: $e^{\mu}$ is the temperature at which 50% of the permafrost is thawed, while $\sigma$ controls how sudden the thaw is around the mean relative to lower and higher temperatures. Technically, permafrost area could increase in the case of cooling temperatures, and therefore the area fraction could be greater than one. However, because even the most aggressive climate action scenarios show future temperatures that stabilize above early 21[st] century temperatures, we assume that permafrost area will never grow more than the starting value.

The lognormal CDF was chosen for several reasons. Its curvature captures the "activation energy" of permafrost thaw with respect to temperature for low temperature change (left side of the curve), and, more importantly, the "diminishing returns" of permafrost thaw at higher temperatures because the more accessible near-surface permafrost has already thawed by that point. Additionally, its parameters are readily interpretable in terms of the timing of 50% permafrost loss ($e^\mu$) and the rate of permafrost loss around the 50% point relative to earlier/later in the process ($\sigma$), which facilitates the use of this framework

to emulate global permafrost dynamics in more complex models. Finally, it is naturally bounded between 0 and 1, which is appropriate as a model of the remaining permafrost fraction.

     While our tuned lognormal CDF aligns well with previous model results (see Section 2.3), there are a variety of possible choices for this functional form and others can be explored in future model development efforts. Fortunately, the modular design and coding best practices of Hector make it simple to substitute alternatives for this equation.

**2.1.1   Permafrost Carbon Emissions**

Even after thaw, only a fraction of permafrost carbon is available for decomposition. While in reality turnover times of soil organic carbon fall anywhere along the range from a few days to thousands of years (Schädel et al., 2014), we group soil decomposition broadly into labile and non-labile pools, where carbon in the non-labile (static) pool decomposes on the order of up to thousands of years and is assumed to be inert for the purpose of this analysis. In Hector, a static fraction of total thawed

permafrost carbon, $f_{static}$, is used to determine a separately tracked value of the total static carbon within the thawed permafrost carbon pool ($static_c$) at each time step before decomposition. For group $i$ at time $t$ for all time steps where $\Delta C_{perm}[i]$ is positive (permafrost is thawing),

$$static_c[i,t] = static_c[i,t-1] + f_{static} * \Delta C_{perm}[i,t] \tag{12}$$

     In the case of refreeze, carbon is removed from $static_c$ proportional to the amount of static carbon currently in the thawed

permafrost pool. In the interest of computational efficiency, this value is not included as a separate carbon pool in Hector, but rather is simply a variable to track the amount of static carbon within the thawed pool over time.

     Of the remaining labile carbon in the thawed carbon pool, most decomposes aerobically to $CO_2$ from microbial respiration, while a small fraction generates $CH_4$ emissions from anaerobic respiration. Heterotrophic respiration emissions from Hector's thawed permafrost carbon pool are partitioned between $CO_2$ and $CH_4$ based on a $CH_4$ respiration fraction, $f_{CH4}$.

With the addition of permafrost in Hector, the total heterotrophic respiration flux of $CO_2$ ($RH[i,t]$) for group $i$ at time $t$ is the sum of heterotrophic respiration in detritus ($RH_d$), soil ($RH_s$), and thawed permafrost ($RH_{pf}$):

$$RH[i,t] = RH_s[i,t] + RH_d[i,t] + RH_{pf}[i,t] \tag{13}$$

     The thawed permafrost $CO_2$ respiration flux, $RH_{pf}$, is proportional to the size of the thawed pool, $C_{thawed}$, based on the static fraction of carbon in that pool, $f_{static}$, and to the fraction of emissions released as $CH_4$, and increases exponentially with

the 200-year running mean of temperature, following the formulation from Hector's default soil pool.

$$RH_{pf}[i,t] = (1 - f_{CH4}) \cdot (C_{thawed} - static_c) \cdot Q_{10}[i]^{T_{200}[i,t]/10} \tag{14}$$

The CH$_4$ respiration flux from thawed permafrost is estimated similarly, but is added to natural CH$_4$ emissions in Hector, which are prescribed at 300 Tg year$^{-1}$ (Hartin et al., 2015) to affect atmospheric CH$_4$ concentrations.

$$RH_{CH4}[i,t] = (f_{CH4}) \cdot (C_{thawed} - static_c) \cdot Q_{10}[i]^{T_{200}[i,t]/10} \tag{15}$$

The total flux of carbon to the atmosphere from the thawed permafrost pool, $F_{thawed-atm}$, is thus:

$$F_{thawed-atm}[i,t] = RH_{CH4}[i,t] + RH_{pf}[i,t] \tag{16}$$

While there are other processes occurring (see Discussion) these are thought to be the major processes controlling decadal permafrost dynamics (Schuur et al., 2015).

## 2.2    Coupled Model Intercomparison Project Data

We used data from the sixth Coupled Model Intercomparison Project (CMIP6) to derive vegetation and litter parameters for the permafrost region as well as to validate our permafrost-temperature curve. Following Burke et al. (2020) we include permafrost grid cells above 20°N that are not covered by ice at the start of the historical period. Permafrost is defined by grid cells where the two-year mean soil temperature at the depth of zero annual amplitude ($D_{zaa}$) of ground temperature remains below 0°C for at least two years. In models where the maximum soil depth is less than the $D_{zaa}$, temperature in the deepest available soil 175 layer was used. This approximation may result in somewhat underestimating permafrost extent. High latitude temperatures and permafrost vegetation and litter values were estimated by masking out non-permafrost grid cells.

We chose models used in Burke et al. (2020), but several of these models did not report the necessary variables in the Earth System Grid Federation archive, so we used only ACCESS-ESM1-5, CNRM-ESM2-1, CanESM5, GISS-E2-1-G, MIROC6, MPI-ESM1-2-HR, MRI-ESM2-0, and NorESM2-LM for comparing our permafrost-temperature relationship (Figure 2b) and 180 our thaw estimates. Of those models only NorESM2, CNRM-ESM2-1, ACCESS-ESM1-5, and CanESM5 reported the relevant carbon outputs and were able to be used in estimating vegetation and litter in the permafrost region.

## 2.3    Configuration and Tuning

To run Hector with permafrost we separated the land component of the model into permafrost and non-permafrost groups, more intuitively thought of as regions in this context. In the permafrost region all parameters were set to the values given in Table 1, 185 and we allocated 3% of the initial global vegetation carbon (equivalent to 17 Pg C) and 11% of the initial detritus carbon (6.1 Pg C) based on the mean share of vegetation and litter carbon in permafrost-containing grid cells in CMIP6 models at the end of the historical simulation. For the fraction of non-permafrost soil carbon in the permafrost region we used a value of 13% of the global non-permafrost soil carbon (equivalent to 308 Pg C, following Hugelius et al. 2014). Initial permafrost carbon in Hector was set to 865 ($\pm$ 125) Pg C based on the 727 Pg C estimate for near-surface (<3 m depth) permafrost by Hugelius 190 et al. (2014) and scaled up based on historical thaw from Koven et al. (2013) so that the resulting modern value is close to 727 Pg C. We did not use the full 1035 Pg C reported in Hugelius et al. (2014) here, as this includes both frozen and non-frozen soil, and we instead allocated the remaining 308 Pg C to non-permafrost soil in the permafrost region.

**Table 1.** Hector configuration of permafrost-related parameters and initial values based on literature review. Ranges shown are used for the sensitivity analysis. $C_{perm}(t=0)$ was estimated by scaling up 727 Pg C (Hugelius et al., 2014) based on the fraction of permafrost thaw in CMIP models (Koven et al., 2013). The soil, vegetation, and litter carbon initial values comprise the non-permafrost carbon pools in the permafrost region, and were estimated from CMIP6 model data and Hugelius et al. (2014). The permafrost thaw parameters $\mu$ and $\sigma$ are tuned parameters, estimated by optimizing the model against results from Koven et al. (2013) while keeping within the upper and lower bounds from Kessler (2017).

| Parameter | Hector Nomenclature | Value | Estimated Range | Reference | Description |
|---|---|---|---|---|---|
| $\mu$ | pf_mu | 1.67 | 1.43-1.91 | tuned to Kessler (2017) | Permafrost-thaw parameter |
| $\sigma$ | pf_sigma | 0.99 | 0.86-1.11 | tuned to Kessler (2017) | Permafrost-thaw parameter |
| $f_{static}$ | fpf_static | 0.74 | 0.4-0.97 | Burke et al. (2012, 2013); Schädel et al. (2014) | static permafrost fraction |
| $C_{perm}(t=0)$ | permafrost_c | 865 Pg C | 740-991 Pg C | estimated from Hugelius et al. (2014) | Initial permafrost carbon |
| $C_{soil}(t=0)$ | soil_c | 308 Pg C | 263-352 Pg C | Hugelius et al. (2014) | Initial non-permafrost soil C in the permafrost region |
| $C_{veg}(t=0)$ | veg_c | 16.5 Pg C | 3.17-29.8 Pg C | derived from CMIP6 model data | Initial vegetation C stock in the permafrost region |
| $C_{litter}(t=0)$ | litter_c | 6.06 Pg C | 1.24-10.9 Pg C | derived from CMIP6 model data | Initial detritus C stock in the permafrost region |
| $wf$ | warmingfactor | 2.0 | 1.75-2.25 | Pörtner et al. (2019) | High-latitude warming factor |
| $f_{CH4}$ | rh_ch4_frac | 0.023 | 0.006-0.04 | Schuur et al. (2013); Nzotungicimpaye and Zickfeld (2017); Schädel et al. (2016) | Fraction of thawed permafrost carbon decomposed as CH$_4$ |

We also amplified warming in the permafrost region as a constant multiple of global mean temperatures in Hector, to account for increased rates of warming at high latitudes. We set this warming factor, $wf_{HL}$, to 2.0 (Pörtner et al., 2019).

We used the upper and lower bounds ($\pm$ one standard error from the best estimate in Kessler) to recalibrate the model in Kessler (2017) to high latitude temperatures and then fitted our lognormal distribution parameters $\mu$ and $\sigma$ to the upper and lower bounds of this adjusted model version. We then used these parameter ranges to tune the permafrost module against CMIP5 multi-model mean output, using the "L-BFGS-B" method from the optim function in the R stats package. We tuned based on the fraction of permafrost remaining over 1850 to 2005 and from 2005-2100 in RCP4.5 and RCP8.5, as reported in Koven et al. (2013). Our tuned permafrost thaw-temperature relationship aligns well with previous analyses and CMIP6 data (Figure 2). We note that thaw fractions derived from our analysis of CMIP6 model results are not substantially different from

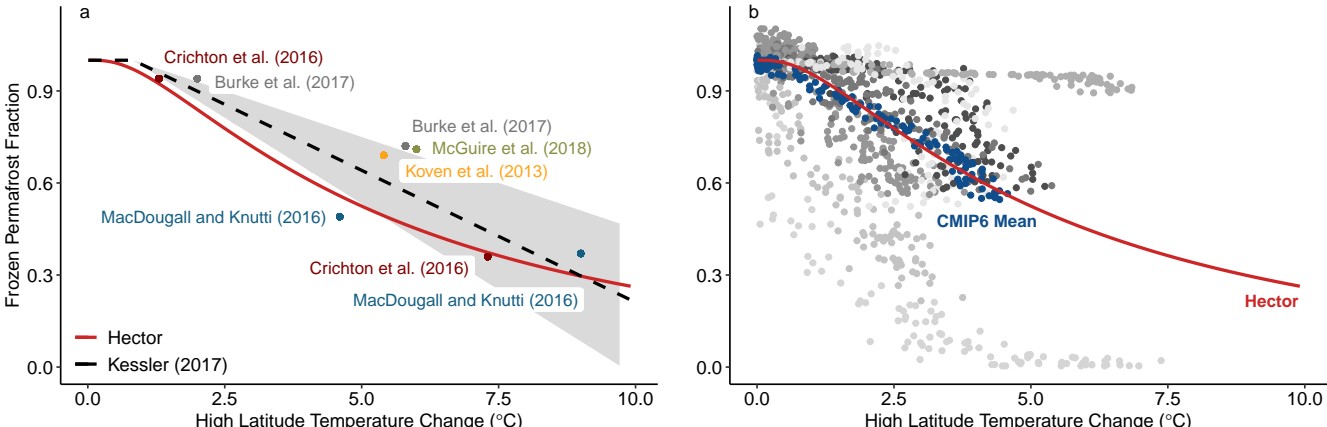

**Figure 2.** a) Lognormal permafrost-temperature relationship (red) in Hector with $\mu = 1.67$ ($e^{\mu}$=5.3) and $\sigma = 0.986$, compared with our high latitude temperature-adjusted form of the linear model in Kessler (2017) (black). The shaded area shows the upper and lower bounds given by plus or minus one standard deviation from our adjusted version of the best estimate model in Kessler. Additional labeled points show results from previous modeling studies for comparison. b) Hector permafrost-temperature relationship (red) shown against CMIP6 data from individual models (shades of gray) and the mean of the models shown (blue).

**Table 2.** Values used for tuning Hector's parameters (column 4) compared against results from Hector after tuning (column 5). The modern permafrost value in Hector was taken from the year 2010. Koven et al. (2013) values are from the top 50% of CMIP5 models reported in that analysis based on accuracy of modern permafrost area. As we do not consider deep permafrost in the model, values for the remaining permafrost area in each time period only include permafrost at less than 3 m depth.

| Scenario | Source | Variable | Value | Hector |
|----------|--------|----------|-------|--------|
| — | Hugelius et al. (2014) | Modern Permafrost Carbon 0-3m (Pg C) | 727 | 730 |
| RCP4.5 | Koven et al. (2013) | Remaining Permafrost Area 1850-2005 (%) | 84 | 85 |
| RCP4.5 | Koven et al. (2013) | Remaining Permafrost Area 2005-2100 (%) | 58 | 56 |
| RCP8.5 | Koven et al. (2013) | Remaining Permafrost Area 2005-2100 (%) | 29 | 32 |

CMIP5, as also found by Burke et al. (2020), and tuning to these instead affected our permafrost thaw parameter values by less than 0.1%.

Our tuned model results closely aligned with the findings in Koven et al. (2013) and gave us a modern permafrost carbon value very close to that in Hugelius et al. (2014) (Table 2). The final tuned value of $\sigma$ that we used as our default baseline in this analysis was 0.986, while the tuned value of $\mu$ was 1.67, which is at the lowest end of the range we used for tuning. To give a more intuitive sense of this number, $e^{\mu}$, or 5.3°C, corresponds to the high latitude temperature difference since pre-industrial at which only 50% of all shallow permafrost will remain.

Estimates of the fraction of static carbon (not vulnerable to decomposition) vary widely and still have a high uncertainty (Kuhry et al., 2020), but we use a mean of 0.74 (0.4-0.97) based on estimates by Schädel et al. (2014) with the upper bound derived from the same analysis and a lower bound from the best estimate given in earlier work by Burke et al. (2012, 2013), which overall found a far smaller static fraction.

The partitioning between $CH_4$ and $CO_2$ emissions from thawed permafrost carbon systems has limited estimates available in the literature (Dean et al., 2018) and is fairly uncertain (Schädel et al., 2016; Schuur et al., 2013). It also depends on soil drainage and anoxia, neither of which are explicitly modeled in Hector, and it may be substantially affected by abrupt thaw processes Dean et al. (2018); Turetsky et al. (2020). For our default parameterization, we set the share of $CH_4$ to be 2.3% (0.6% - 4%) of total emissions. The default value we chose is based on expert assessment in Schuur et al. (2013), and the range is from a meta-analysis of incubation data (Schädel et al., 2016) and a recent review on the contribution of $CH_4$ to the permafrost feedback (Nzotungicimpaye and Zickfeld, 2017). While the $CH_4$ fraction is also known to vary with temperature (Yvon-Durocher et al., 2014), we make the simplifying assumption that the $CH_4$ fraction of overall emissions is static over time. As further estimates of this relationship are published, we can update our model parameterization.

## 2.4 Evaluation

We ran Hector with and without permafrost feedbacks using forcings from each of four Representative Concentration Pathways (RCPs), RCP2.6, RCP4.5, RCP6.0, and RCP8.5 (Moss et al., 2010). We chose these scenarios to broadly demonstrate the impacts of a wide range of future climate conditions on permafrost thaw and permafrost-driven carbon emissions and for ease of comparison with other results. The only difference between our model runs with and without permafrost feedbacks is that the baseline (no-permafrost) configuration of Hector is initialized with $C_{perm}(t = 0)$ set to 0 to turn off permafrost feedbacks. Our analysis focused on the 21[st] century, but we also show some longer term effects of permafrost out to 2300. Hector has not been calibrated over this period, however, and these findings should be taken as provisional. We also ran the model with and without active $CH_4$ emissions to estimate the separate contributions of permafrost-driven $CO_2$ and $CH_4$ emissions to the permafrost climate feedback.

Given that much uncertainty remains surrounding permafrost controls, we evaluated the sensitivity of the model to changes in several of the permafrost-specific controls available in Hector across their estimated ranges from the literature (Table 1). The parameters we include are the permafrost thaw parameters, $\mu$ and $\sigma$; the initial size of the shallow permafrost pool available for thaw ($C_{perm}(t = 0)$); the fraction of thawed permafrost that is not available for decomposition ($f_{static}$); the warming factor used in the permafrost region ($wf_{HL}$), and the fraction of thawed permafrost carbon emissions that decomposes to $CH_4$ ($f_{CH4}$). We additionally include a combined value of the total non-permafrost carbon ($nonpf_c$) in the permafrost region across the soil, vegetation, and litter pools. The respective fractions of each pool are derived for each value of $nonpf_c$ based on a linear fit of their mean, upper, and lower bound shares.

We generated priors for our sensitivity analysis using normal distributions centered on the default values of each parameter from Table 1 with standard deviations taken as the mean difference between the default value and the upper and lower bounds. We then ran Hector with 500 parameter sets randomly sampled from the prior distributions and forced with RCP4.5 emissions.

We focused on three key climate and carbon cycle outcomes: temperature anomalies and atmospheric $CO_2$ and $CH_4$ concentrations. Based on the effects on each outcome in 2100, we estimated the coefficient of variation, elasticity, and partial variance of each parameter.

Briefly, the coefficient of variation describes the uncertainty in the parameter (calculated as the parameter variance divided by the mean), the elasticity describes the sensitivity of the model to a relative change in the parameter, and the partial variance synthesizes these two metrics to describe the relative contribution of uncertainty in a parameter to the total predictive uncertainty in the model output (i.e., the parameters that have the highest partial variance are those that are highly uncertain and to which the model is highly sensitive; parameters that are highly uncertain but to which the model is relatively uncertain, and conversely, parameters to which a model is highly sensitive but whose values are known precisely, would both have low partial variance).

We generally followed the approach of LeBauer et al. (2013), which sampled from parameter distributions to generate an ensemble of model runs that approximate the posterior distribution of model output that can be used in the sensitivity analysis. The sensitivity analysis is based on univariate perturbations of each parameter of interest, and the relationship between each parameter and model output is approximated by a natural cubic spline. The model sensitivity is then based on the derivative of the spline at the parameter median. In our analysis, instead of a cubic spline, we used a multivariate generalized additive model regression. This allowed us to calculate partial derivatives across the median of each parameter, making for simpler computation and easier interpretation.

We also visualized the sensitivity of the model to parameter changes more concretely by estimating temperature sensitivity in Hector to unit changes in each parameter over this century, and the net effect on temperature in 2100 of varying each parameter across its full range (Table 1) in all RCPs. This was estimated by running Hector with parameter values uniformly sampled across each parameter's range while holding all other parameters at their default values. This neglects potential interactive effects, but nonetheless provides useful insights about the impact of our parameter choices and their uncertainty on our results.

## 3 Results

This Hector implementation of permafrost thaw and loss reproduced the magnitude and general temporal trajectory of globally averaged permafrost thaw simulated by ESMs and by simpler permafrost thaw models (Koven et al., 2015a; Burke et al., 2017; Schuur et al., 2015; McGuire et al., 2018). In RCP 4.5, 6.0, and 8.5, permafrost losses, including both thawed permafrost and permafrost carbon that has been decomposed and emitted to the atmosphere, reached 350-450 Pg C by 2100, with the rate of thaw fastest over the 21$^{st}$ century and slowing thereafter (Figure 3a). RCP2.6 is unique in that strong emissions mitigation in this scenario led to cooling temperatures, which allowed for permafrost recovery (i.e., re-freeze of carbon from the thawed permafrost pool) to begin by the end of the century in Hector. In all scenarios, the thawed permafrost carbon pool increased to a peak between the middle and the end of the 21$^{st}$, after which losses to $CH_4$ and $CO_2$ from heterotrophic respiration began to outpace the carbon inputs from new permafrost thaw. Thawed permafrost carbon stocks were limited in their ability to

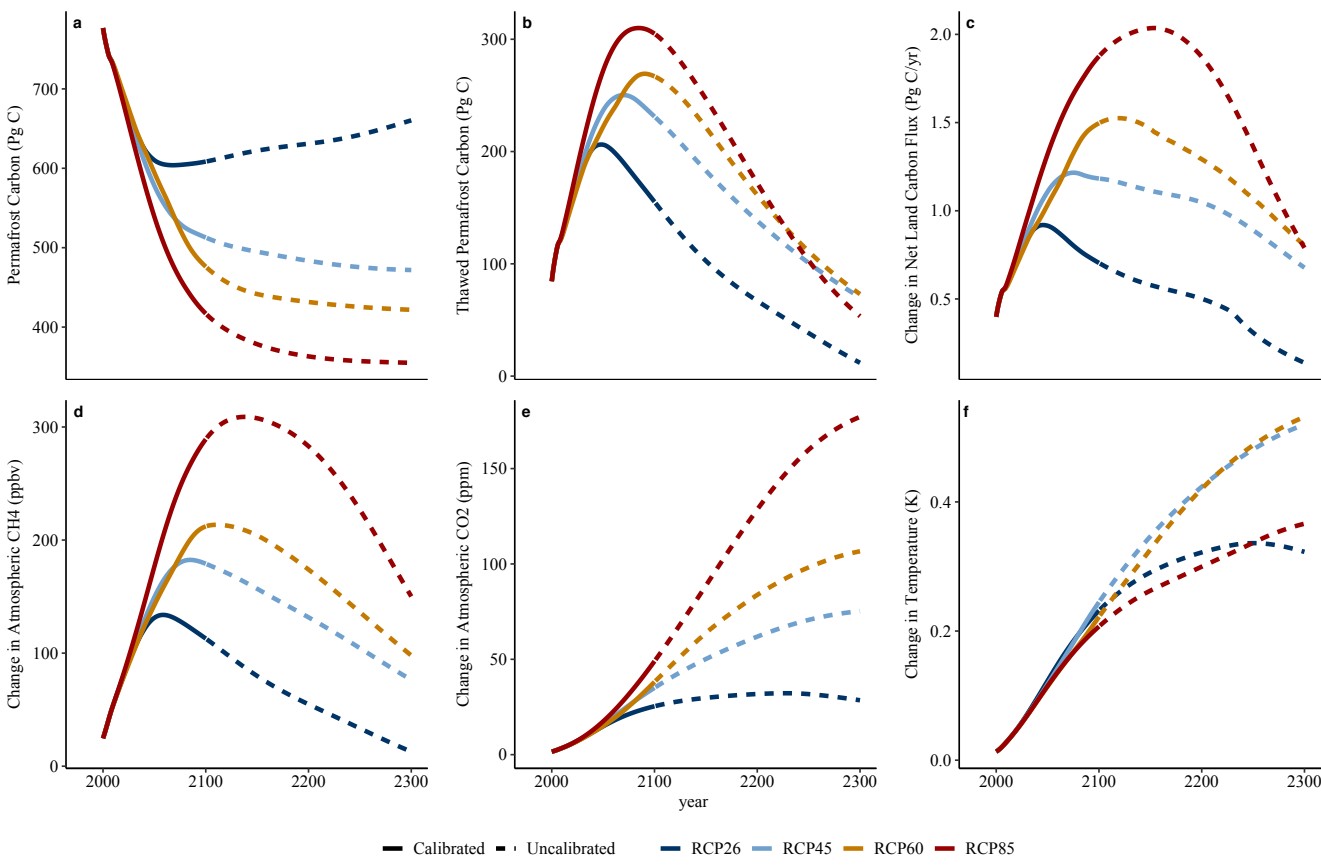

**Figure 3.** Effect on key climate and carbon outputs of including permafrost in Hector, shown as the difference between a model run with and without active permafrost processes under the default model configuration across RCP2.6, RCP4.5, RCP6.0, and RCP8.5. Results are shown through 2100 (solid lines) as the calibrated period of Hector, but are extended to 2300 (dashed lines) to illustrate potential long term dynamics. The net land carbon flux is the sum of the land-atmosphere carbon fluxes: soil, detritus, and thawed permafrost respiration fluxes of $CO_2$, thawed permafrost $CH_4$ emissions, land use change, and net primary productivity, and is defined as positive into the atmosphere.

decompose fully over longer timescales by the labile fraction, though in RCP 2.6 refreeze removed static and labile carbon alike from this pool.

The influence of permafrost on the net land-atmosphere carbon flux in Hector was strongest while respiration emissions from permafrost thaw were at their peak, after 2100, resulting in a maximum increase of around 2 Pg C yr$^{-1}$, somewhat higher than previous findings in Burke et al. (2017) which showed a peak increase of between 1 and 1.5 Pg yr$^{-1}$ in RCP8.5, and closer to 0 280 in RCP 4.5 and RCP2.6. This somewhat offset the existing land sink over the 21$^{st}$ century, reducing it by between 30 and 60%. By 2300, the influence of permafrost on this flux had dropped to closer to 1 Pg C yr$^{-1}$ (Figure 3c). The inclusion of permafrost in the model had almost no effect on the land-atmosphere flux purely from non-permafrost C pools.

We found that including $CH_4$ emissions (set to the default fraction of 2.3% of emissions) in the model resulted in a 24-29% increase in the effect of the permafrost feedback on global mean temperatures, adding around 0.06 °C of warming by 2100 across the RCPs. The relatively short lifetime of $CH_4$ in the atmosphere (estimated as 9.1 years by Stocker et al., 2013) means that the effects of the permafrost carbon feedback on atmospheric $CH_4$ concentrations across the RCPs followed a similar trajectory to that of thawed permafrost carbon, though lagged by several years. As the thawed permafrost carbon pool shrank and $CH_4$ emissions from this pool declined, permafrost-driven changes in atmospheric $CH_4$ also dropped off over the $22^{nd}$ and $23^{rd}$ centuries (Figure 3b,d). The much longer lifetime of atmospheric $CO_2$ (300 to 1000 years; Stocker et al. 2013), meant that the permafrost-driven increases remained over the entire model run time, long after emissions from the thawed permafrost began to decline. By 2100, permafrost emissions increased atmospheric $CO_2$ by between 25 and 50 ppm across all RCPs, and by 2300, in all but RCP2.6, the permafrost-driven increase in $CO_2$ concentrations had substantially grown to between 75 and 177 ppm.

Permafrost emissions also drove a steady increase in temperature over the $21^{st}$ century, continuing to increase through 2300, again in all scenarios but RCP2.6. Consistent with previous findings (e.g., Burke et al., 2017; MacDougall et al., 2012, 2013), the influence of permafrost on temperature resulted in relatively similar effects on absolute temperatures across all RCPs this century (Figure 3f, an increase of between 0.2 and 0.24 °C by 2100). This meant that the effect was relatively less significant in higher emissions scenarios, declining from a 15% increase in RCP2.6 to a 4% increase in RCP8.5 at 2100 (Table 3). Over longer timescales the temperature effects grow more distinct by scenario; the highest absolute permafrost-driven increases in warming were in RCP4.5 and RCP6.0 (0.52 and 0.53 °C in 2300), leaving RCP8.5 as only the third highest beyond 2250 (Figure 3f), although total temperature change in Hector was still highest in RCP8.5. This is due to reductions in the effect of additional carbon emissions on radiative forcing at higher atmospheric carbon concentrations in the model (Hartin et al., 2015). These temperature changes found by our model are similar to those in several previous studies (MacDougall et al., 2012; Burke et al., 2017) (see Section 4.2).

## 3.1 Permafrost Effects on Carbon Pools

Across the four RCP scenarios, between 259 and 458 Pg C (in RCP2.6 and RCP8.5, respectively) of permafrost carbon was thawed by 2100 when all permafrost parameters were set to their default values from Table 1. Between 2000 and 2100 this newly available carbon moved from the thawed pool to the atmosphere and then into the ocean and non-permafrost land carbon pools (Figure 4). In RCP8.5 32% (146 Pg C) was decomposed and emitted to the atmosphere as $CO_2$ and $CH_4$ by the end of the century. Of that 32%, around 100 Pg C remained in the atmosphere, while 23 Pg C was taken up by the ocean and 6 and 8 Pg C respectively were taken up by the non-permafrost soil and vegetation pools. The effect on the detritus pool was less than 1 Pg C. Over longer timescales, the fraction of thawed permafrost carbon emitted to the atmosphere through respiration grew to nearly 90% by 2300, though similar proportions of the permafrost-driven carbon release (here including both permafrost carbon and net carbon losses from non-permafrost soils) were taken up by Hector's other carbon pools. The higher temperatures also drove net losses in non-permafrost soil carbon by 2300 relative to a model run without permafrost, which is included here with the

**Table 3.** Permafrost results across all RCP scenarios at 2100 for several key carbon and climate outputs. All results are global and summed across permafrost and non-permafrost regions. The 'total' columns are generated by running Hector with the configuration in Table 1, while the 'change' columns give the percent change from a baseline model run without active permafrost.

| | Scenario | | | | | | | |
| --- | --- | --- | --- | --- | --- | --- | --- | --- |
| | RCP26 | | RCP45 | | RCP60 | | RCP85 | |
| Output | Total | Change (%) | Total | Change (%) | Total | Change (%) | Total | Change (%) |
| Permafrost Carbon (Pg C) | 608.5 | $-26.2$ | 512.8 | $-37.8$ | 476.1 | $-42.3$ | 417.0 | $-49.5$ |
| Net Permafrost $CO_2$ Emissions (Pg C) | 100.9 | 100.0 | 120.6 | 100.0 | 121.6 | 100.0 | 142.3 | 100.0 |
| Change in Atmospheric $CO_2$ (ppm) | 408.4 | 6.6 | 539.4 | 6.9 | 686.8 | 5.9 | 943.8 | 5.5 |
| Net Permafrost $CH_4$ Emissions (Pg C) | 2.4 | 100.0 | 2.8 | 100.0 | 2.9 | 100.0 | 3.4 | 100.0 |
| Change in Atmospheric $CH_4$ (ppbv) | 1300.1 | 9.5 | 1841.5 | 10.8 | 2000.1 | 11.9 | 4581.5 | 6.7 |
| Non-Permafrost Soil Carbon (Pg C) | 1856.6 | 0.6 | 1916.9 | 0.5 | 1952.0 | 0.4 | 1960.9 | 0.3 |
| Detritus Carbon (Pg C) | 60.6 | 1.3 | 63.8 | 1.1 | 66.4 | 0.8 | 68.5 | 0.6 |
| Vegetation Carbon (Pg C) | 571.5 | 1.6 | 608.6 | 1.5 | 629.8 | 1.2 | 667.7 | 1.2 |
| Temperature Anomaly (°C) | 1.8 | 14.5 | 2.8 | 9.5 | 3.4 | 7.0 | 4.9 | 4.4 |

permafrost carbon in the calculations involving non-permafrost carbon pools as Hector does not currently have a meaningful way to evaluate carbon sources within a pool (Figure 4).

While scenarios with lower radiative forcing thawed less permafrost carbon overall, a somewhat higher fraction of that carbon ended up released into the atmosphere (40% by 2100 and 94% by 2300 in RCP2.6). Relatively more of the permafrost-driven carbon release was also taken up by the ocean in this scenario (26% by 2100 and nearly 60% by 2300) thanks to lower mean global temperatures increasing the solubility of $CO_2$ in seawater, while 53% (54 Pg C) remained in the atmosphere by 2100 (31% by 2300; Figure 4).

## 3.2 Model Sensitivity to Permafrost Parameters

Based on the effects on end-of-century temperature change and atmospheric $CO_2$ and $CH_4$ concentrations, we found that the most significant permafrost control in Hector was the static fraction, which supports similar findings by previous studies (Koven et al., 2015a; MacDougall and Knutti, 2016). This accounted for 68% of the partial variance in temperature (around 30% in $CH_4$, and 72% in $CO_2$) across all three outcomes (Figure 5). The second most significant parameter in terms of temperature was the initial permafrost carbon value which accounted for 10% of the partial variance, followed by the mean thaw parameter ($mu$, 9%). The $CH_4$ fraction and high latitude warming factor had small effects (6 and 7%, respectively), while varying the standard deviation thaw parameter ($\sigma$) and the initial non-permafrost carbon in the permafrost region across their ranges had almost no impact on any output variable. The effect of the $CH_4$ fraction was much more significant in terms of its effects on atmospheric $CH_4$ (59%), but had no discernible effect on $CO_2$ concentrations. Over longer timescales (out to 2300) the influence of the

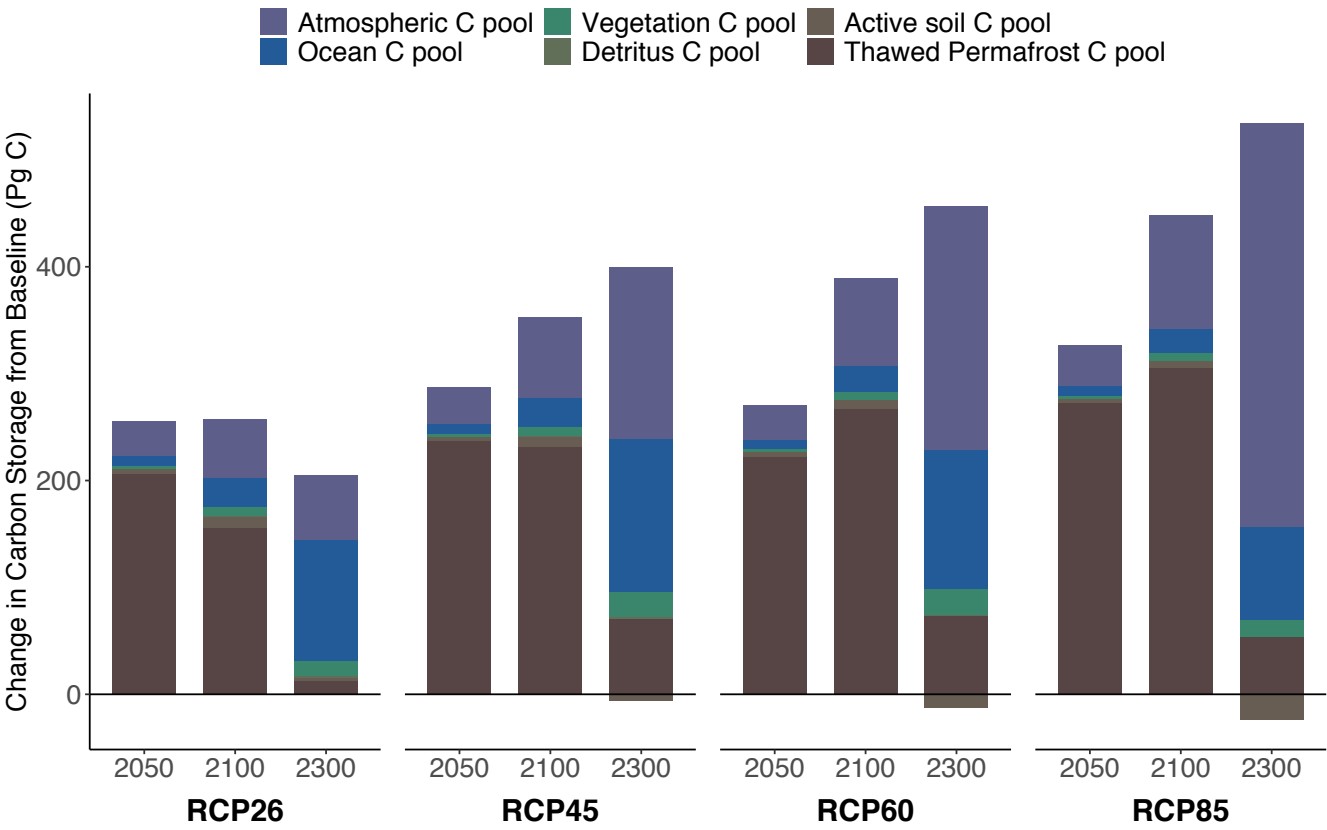

**Figure 4.** Changes in carbon stocks in a permafrost-active model run compared to a run without permafrost at 2050, 2100, and 2300 across all RCPs. The sum of each bar is the total carbon lost from the permafrost pool by that year in each RCP. Results for 2300 should be taken as provisional since Hector is not calibrated over this period. While more carbon moves from the thawed pool into the atmosphere, and then into the ocean across the three periods shown, a relatively larger fraction of carbon remains in the atmosphere in higher warming scenarios.

warming factor increased somewhat, while the influence of the $CH_4$ fraction on temperature decreased to nearly zero, which follows from the decline in permafrost-driven changes in atmospheric $CH_4$ by this time (Figure 3).

The temperature response of the model to a unit increase in each parameter generally strengthened over time, with the exception of the permafrost thaw parameter $\sigma$ which had a larger impact early on before declining to a sensitivity of 0.006 °C $10\%^{-1}$ (Figure 6a). Varying the static fraction caused the strongest temperature response, a ~0.04 °C decrease in temperature for every 10 % increase in $f_{static}$ at 2100. The permafrost thaw parameter $mu$ had the next strongest sensitivity by the end of this century, -0.03 °C $10\%^{-1}$, and also varied the most across the RCPs. Temperature exhibited the strongest positive

sensitivity to changes in the high latitude warming factor and initial size of the permafrost carbon pool (0.03 °C $10\%^{-1}$ and 0.02 °C $10\%^{-1}$, respectively).

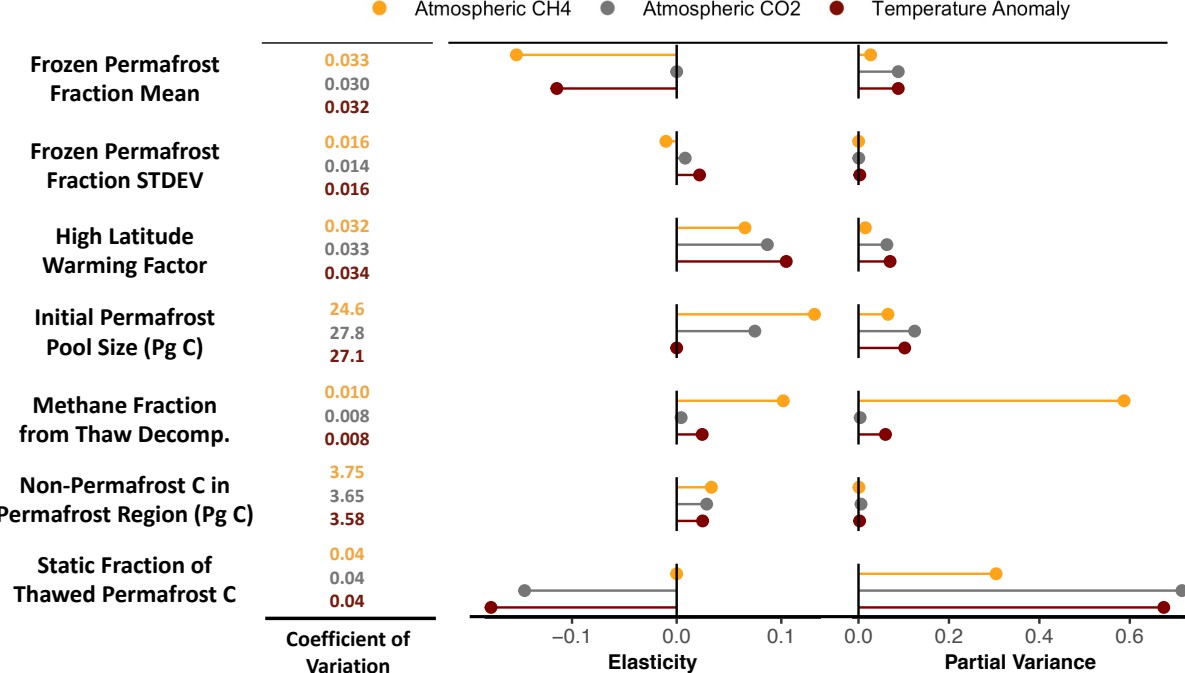

**Figure 5.** Sensitivity analysis of the effect of key permafrost controls on end-of-the-century atmospheric $CH_4$ (orange) and $CO_2$ (gray) concentrations as well as temperature anomalies (dark red), following LeBauer et al. (2013) and forced with RCP4.5 emissions. The coefficient of variation is the ratio between the input parameter mean and variance and reflects the parameter's relative uncertainty, elasticity is the normalized sensitivity of the model to a change in a particular parameter, and the partial variance, or the fraction of variance in the model output that is explained by the given parameter, integrates the elasticity and coefficient of variation to give the overall sensitivity of the model to each parameter.

In practical terms, the effects of varying the static fraction over its plausible range (Table 1) on permafrost-driven temperature change spanned nearly 0.4 °C by 2100 across all RCPs, or up to a 0.2 °C impact compared to the default value (Figure 6b). At the extremes of their potential ranges, the permafrost thaw parameter $mu$, the high latitude warming factor, the initial size of the permafrost pool, and the $CH_4$ fraction each had net effects of between +0.04 and +0.06 °C compared to a run at their default values. Consistent with our findings in Figure 5, the non-permafrost carbon and permafrost thaw parameter $\sigma$ had only a minimal impact on temperature when varied over their ranges, around 0.01 °C each.

## 4   Discussion and Conclusions

Including permafrost in Hector significantly increased end-of-century atmospheric $CO_2$, $CH_4$, and warming, though the impact on atmospheric $CH_4$ was declining somewhat by the end of the model run. The parameter with the most significant effects on

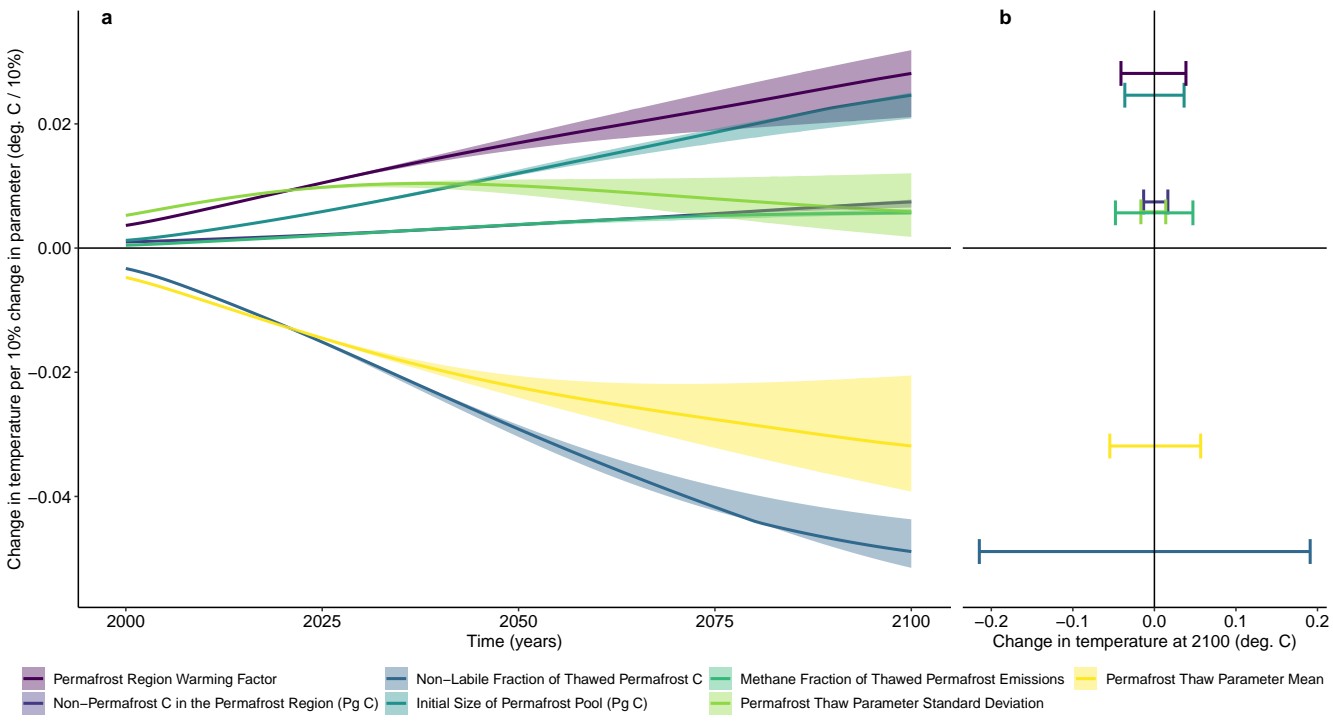

**Figure 6.** Sensitivity of temperature over the 21[st] century across RCP2.6, RCP4.5, RCP6.0, and RCP8.5 to variations in each of the key permafrost parameters in the model. Panel a) shows the sensitivity of temperature in Hector to unit changes in each parameter from its default value, and how that sensitivity varies over time and by emissions scenario. Shaded regions correspond to the range across RCP2.6, RCP4.5, and RCP8.5, while the solid line shows the median. Panel b) gives the total effect on temperature in 2100 from varying each parameter across its potential range - in other words, how the potential sensitivities in panel a) translate to practical effects at the end of the century based on the actual ranges of each parameter.

these outcomes was the fraction of permafrost not available for decomposition, or the static fraction. This suggests that further research constraining this parameter continues to be important for reducing uncertainty in permafrost estimations moving forward. While other studies have supported this finding (MacDougall and Knutti, 2016; Koven et al., 2015a), it is still important to acknowledge that the significance of any parameters in Hector is limited by the simplicity of the permafrost representation 355  we are able to include and may change with more detailed, physically-based representations of the processes involved.

### 4.1  Model Limitations

While we attempted to use reasonable values for our model parameters and calibrated Hector to emulate the behavior of permafrost thaw in global climate models, these results should be taken as demonstrative of this model's capabilities, rather than conclusive projections, as model parameter values can be adjusted as needed to reflect the latest understanding of permafrost

characteristics, and this was not our focus here. What is more important is to acknowledge the permafrost dynamics that are not captured in this model's structure.

Hector's permafrost module parameterizes gradual permafrost thaw, following previous development on simple climate models (Kessler, 2017), but leaves off consideration of abrupt thaw, which has been found to be a potentially significant contributor to future permafrost emissions (Turetsky et al., 2020), increasing the overall permafrost soil carbon emissions by
365 125-190% above that from gradual thaw and increasing the contribution of $CH_4$ to those emissions, according to a recent analysis (Anthony et al., 2018). Abrupt thaw is also missing from current Earth system models, so our tuning to these models would not account for this mechanism, and it may mean that Hector is somewhat underestimating the permafrost carbon feedback. Abrupt thaw is also a key process for permafrost in peatland soils, and a recent analysis estimates an additional 40 Pg of permafrost carbon stored in peat than had been found previously (Hugelius et al., 2020). Based on our sensitivity analysis,
increasing the initial permafrost by this amount might translate to around a 0.02°C increase in overall temperature change by 2100.

Thawing permafrost, particularly abrupt thaw processes, can affect geometry and drainage patterns of the landscape, including creating thaw lakes which are persistent sources of both $CH_4$ and $CO_2$ (Vonk et al., 2015; Matveev et al., 2016). Hector does not include hydrological processes nor abrupt thaw mechanisms that could account for this effect, and this additional
375 consequence of permafrost thaw on emissions would not have been captured through tuning to CMIP models because we only tuned Hector against the fraction of permafrost thaw in each. While we found that permafrost emissions from Hector's thawed pool dropped over time as thaw slowed and the thawed pool decomposed, the model is missing this longer-term affect of permafrost thaw on $CH_4$ and $CO_2$ emissions in the region.

The absence of hydrological processes in Hector also means the model misses interactions between permafrost thaw and
380 soil moisture. Soil moisture has been found to play a critical role in the rate of release of thawed permafrost carbon, as drier soils release carbon much faster than wetter soils (Elberling et al., 2013). Thawing permafrost itself impacts soil moisture, although predicting these effects is difficult (Wickland et al., 2006). Moisture also affects the balance of aerobic and anaerobic decomposition, determining the ratio of $CO_2$ to $CH_4$ release (Turetsky et al., 2002). For example, Lawrence et al. (2015) found that permafrost thaw increased soil drying, reducing the $CH_4$ fraction of permafrost emissions to the extent that the global
warming potential of emissions from the permafrost region was reduced by 50%. Projections of drying soils due to permafrost thaw are also supported by the analysis in Andresen et al. (2020).

Hector's permafrost module also only accounts for carbon stored in the top three meters of soil, as this shallow permafrost is the most vulnerable to both thaw and decomposition (Kessler, 2017). However, an analysis accounting for abrupt thaw found higher contributions from deep carbon when including these abrupt thaw processes (Schneider von Deimling et al., 2015;
Anthony et al., 2018). Previous modeling results have found that ~2 Pg C may be emitted over the next century from this deeper permafrost (Koven et al., 2015b), or an additional 3% of total permafrost-driven carbon emissions over that time period, but this study also neglected abrupt thaw processes. There may also be a larger contribution from this pool over longer-term results since warming would have more time to reach these deposits, although warming in Hector levels off beyond the end of the century.

While other mechanisms are included in ESMs and some of their effects on permafrost thaw can be implicitly captured through calibration, not explicitly modeling these effects can still impact temporal dynamics and the relative strength of particular outcomes. A key difference between Hector and ESMs is spatial representation. While ESMs are spatially explicit, Hector is primarily global, although with separate calculations for land regions or other groups. In the case of the results shown here, only a single permafrost category was used; this combines high latitude and high elevation permafrost, although in reality these may be differently affected by climate. Future analyses with this model may choose to further sub-divide the permafrost region into more specific categories to better address these different dynamics.

We also made the simplifying assumption that thawed permafrost carbon does not interact with the vegetation or detritus pools, and that newly thawed permafrost carbon does not affect the potential size of the vegetation and detritus pools in the permafrost region. This means our results exclude any potential changes in plant productivity as a result of permafrost thaw, including any due to changes in nutrient availability, though the sign of these effects is highly uncertain (Frost and Epstein, 2014; Li et al., 2017).

An additional area of focus for future work should be Hector's handling of heterotrophic respiration in soil, which currently uses a fairly arbitrary 200-year running mean of air temperature as a proxy for soil temperature. This controls soil decomposition and thus climate effects in Hector, including from permafrost, and should be further evaluated against alternative functional forms.

Finally, we do not include any insulating effect from snow and vegetation, which can protect permafrost from warmer air temperatures (Shur and Jorgenson, 2007). However, this effect may be small on the global scale, as including such protected permafrost was not found to substantially alter the amount of permafrost thaw over the next century of warming according to a 2017 analysis by Chadburn et al., though this analysis used equilibrium temperatures and does not give us information about the potential for these insulation effects to play a role in mitigating transient thaw.

Of these limitations, we consider the most significant and likely influential on the magnitude of our results to be the lack of abrupt thaw processes, including the effects of abrupt thaw on deeper permafrost carbon. Results from Anthony et al. (2018) suggest our model may be underestimating the permafrost carbon feedback by as much as 20-50%, though there are still only limited estimates of these effects in the literature. The other significant effect on permafrost emissions estimates in Hector is the lack of hydrological processes, which would potentially generate longer term increases in emissions from permafrost thaw due to lake formation. Other mechanisms affecting rates of permafrost thaw are included in CMIP models and thus we expect to have captured the net end-of-century effects of these mechanisms through tuning to CMIP outputs.

## 4.2   Comparison to Previous Work

While our permafrost model is necessarily limited in complexity by Hector's structure and by the need for computational efficiency, we are able to reasonably reproduce previous results from both simple and more sophisticated models (Table 4). The fraction of permafrost remaining in Hector in RCP8.5 by 2100 aligns fairly closely with the results from CMIP6 models estimated by Burke et al. (2020). Even during the uncalibrated period of Hector, the land area of permafrost lost still compares well against estimates from McGuire et al. (2018) in RCP8.5, though not as well in RCP4.5.

**Table 4.** Comparison of Hector's results to values from previous studies. Since Hector does not account for permafrost in terms of area, we estimated the values for comparison to McGuire et al. (2018) based on the fraction of permafrost lost over this time period, multiplied by the initial permafrost area in McGuire et al. (2018).

| Scenario | Source | Variable | Value | Hector |
|---|---|---|---|---|
| RCP8.5 | Burke et al. (2020) | Permafrost Remaining 2005-2100 (%) | 37 | 32 |
| RCP4.5 | McGuire et al. (2018) | Permafrost Lost 2010-2299 (x$10^6$ km$^2$) | 4.1 | 7.4 |
| RCP8.5 | McGuire et al. (2018) | Permafrost Lost 2010-2299 (x$10^6$ km$^2$) | 12.7 | 12.2 |
| RCP4.5 | MacDougall and Knutti (2016) | Cumulative Permafrost $CO_2$ Emissions 1850-2100 (Pg C) | 71 | 121 |
| RCP8.5 | MacDougall and Knutti (2016) | Cumulative Permafrost $CO_2$ Emissions 1850-2100 (Pg C) | 101 | 142 |
| RCP8.5 | Schuur et al. (2015), Koven et al. (2015) | Cumulative Permafrost $CO_2$ Emissions 2010-2100 (Pg C) | 92, 28-113 | 130.9 |
| — | Kirschke et al. (2013) | Permafrost $CH_4$ Flux 2010 (Tg C yr$^{-1}$) | 30 | 20.7 |
| RCP8.5 | Koven et al. (2015) | Permafrost $CH_4$ Flux Change 2010-2100 (Tg C yr$^{-1}$) | 3.97-10.48 | 59 |
| RCP8.5 | Knoblauch et al. (2018) | Relative Mineralization of Permafrost C 2010-2100 (g $CH_4$ kg C$^{-1}$) | 22 | 5.7 |
| RCP8.5 | Crichton et al. (2016), Burke et al. (2017) | Permafrost-Driven Temperature Change by 2100 (%) | 10-40, 0.2-12 | 4.4-14.5 |
| RCP8.5 | MacDougall et al. (2012) | Permafrost-Driven Temperature Change by 2100 (°C) | 0.27 | 0.21 |

Cumulative permafrost $CO_2$ emissions by 2100 were generally higher than previous results in both RCP4.5 and RCP8.5 MacDougall and Knutti (2016); Schuur et al. (2015); Koven et al. (2015b). The values given in Schuur et al. (2015) includes the entire permafrost profile rather than 0-3m as is represented in Hector, which implies an even stronger difference between these results and Hector's.

The modern $CH_4$ flux in Hector was around 30% lower than that found by Kirschke et al. (2013), and Hector's cumulative $CH_4$ emissions from 2010 to 2100, normalized by the initial permafrost pool size, were much lower than a more recent estimate from incubation data (Knoblauch et al., 2018). But the increase in the $CH_4$ flux the by the end of the century was substantially higher in Hector compared to estimates by Koven et al. (2015b). The $CH_4$ contribution to permafrost-driven temperature change estimated by Hector was between 24 and 29%, somewhat higher than the 16% given in Schaefer et al. (2014), but just under the 30-50% range given by the expert assessment in Schuur et al. (2013).

Previous estimates of the temperature amplification of permafrost carbon feedback by the end of the century cover a wide range, from 0.1 to 0.8 °C in MacDougall et al. (2012) with a best estimate of 0.27 °C, from 10-40% of peak temperature change in Crichton et al. (2016), and 0.2 to 12% of peak temperature change in Burke et al. (2017). In Hector, we find a temperature amplification due to permafrost emissions of 4-15%, or around 0.2 °C, by 2100 across all four RCPs (Table 3), which falls close to the best estimate in MacDougall et al. (2012) and somewhat between the ranges of Crichton et al. (2016) and Burke et al. (2017).

## 4.3 Conclusions

The addition of permafrost thaw in Hector provides a useful tool for understanding the potential impact of the permafrost carbon feedback over the next decades and centuries, a particularly important capability in the context of ongoing climate change and uncertain impacts of permafrost thaw. The model's simplicity means that model parameters and structural components alike can easily be adjusted as further studies improve our understanding of permafrost dynamics, and it can cheaply run uncertainty analyses over a wide range of parameter values to account for the remaining gaps in our knowledge of permafrost controls. In the future, Hector's permafrost module can be easily coupled with economic and human systems models like the Global Change Analysis Model (GCAM) to estimate the economic consequences of warming from this feedback and to improve evaluation of climate and energy policy using such models.

## 5 Code availability

The version of Hector used in this analysis is available at https://doi.org/10.5281/zenodo.4876800 and the code used to generate the tables and figures is available at: https://doi.org/10.5281/zenodo.4876812

*Author contributions.* A.N.S. developed an initial version of this model; D.L.W. updated and revised it to the current version under the mentorship of B.B.-L., analyzed results, and performed a sensitivity analysis. D.L.W. wrote the manuscript with contributions from all co-authors.

*Competing interests.* The authors declare that they have no conflict of interest.

*Acknowledgements.* The authors would like to thank Christina Schädel for her valuable feedback and suggestions on this research. This work was based on research supported by the U.S. Department of Energy (DOE), Office of Science, Biological and Environmental Research, as part of the Regional and Global Model Analysis program. This research is also supported in part by US Environmental Protection Agency, under Interagency Agreement DW-089-92459801. The views and opinions expressed are those of the authors and do not necessarily represent the views or policies of the US EPA or other funding organizations. Support for B.K. was provided in part by the National Science Foundation through agreement CBET-1931641, the Indiana University Environmental Resilience Institute, and the Prepared for Environmental Change Grand Challenge initiative. The Pacific Northwest National Laboratory is operated for the US Department of Energy by Battelle Memorial Institute under contract DE-AC05-76RL01830. Support for A.N.S. was provided in part by the NASA Surface Biology and Geology (SBG) mission study.

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
