# Peer review of "A Permafrost Implementation in the Simple Carbon-Climate Model Hector v.2.3pf"

_Geoscientific Model Development, 2020_

## Referee Comment (RC2)

A Permafrost Implementation in the Simple Carbon-Climate Model Hector

This paper discusses the implementation of a simple permafrost carbon module within the Hector simple climate model. The authors talk about uncertainties but the results have very few details of their impact. In particular, the model results have a high permafrost carbon feedback temperature compared to other results in the literature. I think this is fine if the uncertainties are made more high profile throughout the whole document particularly plumes in Figure 3 but also maybe table 3. Q quick thought - do the authors think this high feedback temperature is caused by a relatively large methane contribution?

I think more details and discussion are required about the physical permafrost change parametrisation. This is a key addition to the model and not explored in much detail. I see it is discussed more in Figure 2 but I think it needs to be compared to something other than Kessler. **Why is a volume fraction not used when considering physical permafrost change?** This is much more relate-able to carbon amount.

Please check the signs/definition of fluxes in the equations and their text.
It would be good to have units included for all variables. Also check all acronyms are defined.

In general the paper contains most of the required information but sometimes later than I would like it.

Minor comments:

Line 3 - permafrost C feedback is hardly ever estimated using ESMs currently.
Line 6 - ?? ESM permafrost estimate
Line 10 - 0.5 degree feedback temperature is relatively high.
Line 20 - add Biskaborn reference (https://www.nature.com/articles/s41467-018-08240-4)
Line 27/29 - Burke et al., 2017 (https://bg.copernicus.org/articles/14/3051/2017/bg-14-3051-2017.pdf)
Line 30 - JULES - Joint UK Land Environment Simulator
Line 31 - remove both 'can' s
Line 35 - these more complex models still have missing/incorrectly parametrised processes and would definitely benefit from uncertainty quantification.
Line 39 - remove 'models'
Line 45 - remove 'and'
Line 48 - The paragraph above suggests that these simple climate models do not have good spatio-temporal resolution but then this line suggests that Hector will be used to evaluate regional impacts.
Line 71 - Equation 1 defines a land-to-atmosphere flux but Equation 3 suggests an atmosphere-to-land flux.
Line 71 - I think the last sign in Equation 1 should be + and the definition of FL shoudl specifically state that NPP and Respiration act in opposite directions
Line 73 - FL is the difference between NPP (uptake) and respiration (loss).
Line 78/79 - what are the 1/4 and 1/50 factors for?
Equation 4 has an air temperature change as a power and Equation 5 - has a running mean air temperature as a power? That looks a bit odd?
Line 85 - Where does the 200 years come from? Can this be justified further?
Line 99 - Equation 6/7 - DCperm is added to both Cperm and Cthawed? Surely it should be subtracted from one and added to the other?
Line 99 - What is Fthawed-atm? Is the sign correct?
Line 103 - shouldn't PHI be a volume fraction?
Line 109 - Please give more details on Equation 9 and explore its validity.

Line 139 - please state earlier that the 308 Pg C soil carbon is 'non-permafrost' carbon

Table 1 caption - it says mu and sigma are from Koven et al, but in the Tabel it says they are from Kessler.

Table 1 - no range for Cosoil or Cveg - have the authours checked if the model is snesitive to these?

Table 1/Line 144 - what is the wf used for? I cant see it in any equation? How does this compare to the value in Chadburn et al. 2017 (https://www.nature.com/articles/nclimate3262)

Table 1 - f_RH_CH4 does not match the name in equation 11/12

Line 150 - cite Burke et al. 2020 (https://tc.copernicus.org/articles/14/3155/2020/tc-14-3155-2020.pdf) which shows CMIP6 and CMIP5 are very similar.

Table 2 - how dies this compare to Burke et al. 2020?

Line 155 - why do we expect mu/sigma to fall within the range of Kessler 2017?

Line 159 - this 70 % was not included in the uncertainty range. Any reason?

Line 161 - uncertainty ranges in Table 1 do not reach the 4.3 % suggested in the text. Why not?

Line 178 - check name f_RH_CH4

Line 179 - how were the parameters sampled from prior distribution? Latin Hypercube?

Line 181-184 - please give more details on what these parameters mean and what the apprach of LeBauer is.

Line 188 - these 300-400 Pg C are not yet decomposed so comprise the Cthawed pool?

Line 195 - is there any evidence that the ~3 Pg C /year has been found in other models. This is quie high.

Figure 3 - Again I think the feedback temperature is generally quite high compared with other simulations. It would be good to see the spread introduced by including the parameter uncertainties.

Figure 3 - Please look up the standard colours for the RCP scenarios and use them. It is a littel confusing to have RCP2.6 as red.

Line 221 - Quite a lot of this carbon remains in the atmosphere (expect ~50 %, ~25% to land/~25% to ocean). Is this a function of the model structure?

Figure 5 looks interesting but is not immediately clear. Please include equation symbols in the names. I am not familiar with all of these statistics so a way to highlight the interesting ones and link them to the text would be great.

Line 235 – the 30-45% is 'partial variance'?

Line 266 – Walter-Anthony, 2018

Line 275 – this should be encompassed by the parameter uncertainty.

Line 275 – 280 – this relates to the abrupt thaw processes discussed above and the two discussed together.

Line 304 – Chadburn et al assumed the soil and air temperatures were in equilibrium in their analysis.

How well does the 200 year temperature term represent the thermal inertia of the permafrost?

Section 4.1 probably need to mention nutrient limitation.

Table 4 – add the results of the temperature effect here. I am not sure why some comparisons are in the table and some are in the text.

Line 334 – please define/reference GCAM.

---

## Author Comment (AC1)

**A Permafrost Implementation in the Simple Carbon-Climate Model Hector**

Dawn L. Woodard, Alexey N. Shiklomanov, Ben Kravitz, Corinne Hartin, and Ben Bond-Lamberty

**Author's responses to comments**

**RC1 comments**

*This study projects permafrost thaw and associated GHG emissions using a new representation of permafrost, which was integrated to the global carbon-climate model Hector. The authors use air temperature projections to quantify permafrost thaw. They acknowledge the limitations that come with the use of a simple model such as theirs, and do not over-interpret their model outputs.*

*I enjoyed reading this manuscript and believe it will be of broad interest to the scientific community as well as informative to IAMs. The Discussion section does a great job at documenting the model limitations (as a field ecologist, I appreciate that); I also think the section that compared this study's model outputs with that of other models was beneficial to the reader.*

We appreciate the reviewer's interest and overall positive assessment.

*I'd be interested to see a few more details pertaining to: (1) How is the "static" (non-labile) C fraction in the permafrost determined? (2) Why is the CH4 emission fraction from thawed permafrost set to 2.3% (please add references or some kind of explanation), and how much effect would a lower or higher fraction have on GHG (provide graphs)? (3) How sensitive is the model to the permafrost pool size (provide graphs of projected GHG under different pool sizes)?*

1) We have added the following sentence to better explain where the non-labile fraction comes from: "While in reality turnover times of soil organic carbon fall anywhere along the range from a few days to thousands of years, we group soil decomposition into fast and slow pools, following previous work (Shädel *et al.* 2014), where the slow fraction decomposes on the order of up to thousands of years and is assumed to be inert for the purpose of this analysis."
2) We revised the relevant sentence to read: "We set the share of $CH_4$ to be 2.3% of total emissions, a value derived based on expert assessment in Schuur *et al.* (2014)."
3) We have added the following figure to visualize the sensitivity of the model to the potential ranges of the key parameters in our sensitivity analysis, including the permafrost thaw parameter µ, the non-labile fraction, the methane fraction of heterotrophic respiration, and the initial permafrost carbon fraction. (Note that the exact values on this figure will change in the revised manuscript but we include it here for reference).

[Figure]

*Minor comments:*

*If possible, add Hugelius et al 2020 (PNAS; https://www.pnas.org/content/117/34/20438) to Table 4, and discuss their findings in light of yours.*

As our results do not separate out peatlands, it is not straightforward to directly compare our results to this paper in Table 4, but we have added the following to our discussion addressing the implications of Hugelius *et al.* 2020 for our results.

"Abrupt thaw is also a key process for permafrost in peatland soils, and a recent analysis estimates an additional 40 Pg of permafrost carbon stored in peat than had been found previously (Hugelius *et al.* 2020). Based on our sensitivity analysis, increasing the initial permafrost by this amount might translate to a little over an additional 1% increase in overall temperature change by 2100 in Hector."

*l-39: "models" is there twice*

Thanks for the catch! Corrected.

*l-181: fix the typo in the word "parameter"*

Corrected.

**RC2 comments**

*This paper discusses the implementation of a simple permafrost carbon module within the Hector simple climate model. The authors talk about uncertainties but the results have very few details of their impact.*

We appreciate the reviewer's careful examination of our paper and thoughtful comments here and below.

We have added a figure (new Figure 6 - above) showing the impact of parameter uncertainties on global mean temperature; this output variable is integrative, capturing the effects on other key outcomes as well (permafrost thaw, methane and carbon dioxide emissions). We have also expanded further on uncertainties in the discussion (described further in various responses below).

*In particular, the model results have a high permafrost carbon feedback temperature compared to other results in the literature. I think this is fine if the uncertainties are made more high profile throughout the whole document particularly plumes in Figure 3 but also maybe table 3. Q quick thought - do the authors think this high feedback temperature is caused by a relatively large methane contribution?*

We agree that the temperature response in this model is high compared to many existing estimates. In our revised manuscript we update to a substantially higher value of the non-labile fraction, 0.72 - based on the more recent and comprehensive Shädel *et al.* (2014), compared to the previous value of 0.4, which ends up bringing our temperature response down to between 0.1 and 0.2 ℃, depending on the scenario. We also will tune the model parameterization of permafrost thaw versus temperature to CMIP6 data in the revised manuscript, which will provide a more robust foundation for our default configuration of this model. We note that one of Hector's strengths is in its ability to calibrate to different assumptions and explore their consequences, so that our default configuration should only be seen as a starting point for looking at the effects of permafrost in this model.

We additionally add more discussion about the comparison between our temperature estimates and those from previous results throughout the paper. We think this higher temperature impact is driven by both the inclusion of methane, which not all previous studies have done and is responsible for ~25% of our end-of-century warming, as well as our use of a relatively high warming factor parameter. We have also added the following sentence to compare the estimated methane-driven warming to that in previous studies: "The methane contribution to

permafrost-driven temperature change estimated by Hector was around 25%, higher than the average of previous estimates (16%) found in Schaefer *et al.* (2014), but within the range of the studies included in that analysis."

*I think more details and discussion are required about the physical permafrost change parametrisation. This is a key addition to the model and not explored in much detail. I see it is discussed more in Figure 2 but I think it needs to be compared to something other than Kessler.*

This is a good point, and we have expanded Figure 2 (below) and the discussion to include other projections of high latitude temperature change versus permafrost thaw fraction.

[Figure]

**Why is a volume fraction not used when considering physical permafrost change?** *This is much more relate-able to carbon amount.*

Because of Hector's structure and global spatial scale, the permafrost component is based only on the movement of carbon by mass, with no explicit representation of depth or area, so this is most accurately thought of as a permafrost mass fraction. We have clarified this in the methods section.

*Please check the signs/definition of fluxes in the equations and their text.*

*It would be good to have units included for all variables. Also check all acronyms are defined.*

*In general the paper contains most of the required information but sometimes later than I would like it.*

We have carefully gone through the paper and checked signs and added units and acronym definitions in all the places we found they were missing.

*Minor comments:*
*Line 3 - permafrost C feedback is hardly ever estimated using ESMs currently.*

We have edited this sentence to read: "The resulting climate feedback can be estimated using land surface models, but the high complexity and computational cost of these models make it challenging to use them for estimating uncertainty, exploring novel scenarios, and coupling with other models."

*Line 6 - ?? ESM permafrost estimate*

As we tuned to several different values, we have left the details of these estimates for later in the manuscript in favor of simplicity within the abstract.

*Line 10 - 0.5 degree feedback temperature is relatively high.*

It is, and while diverging from existing estimates is not necessarily cause for concern, some changes we have made to our default parameterization have ended up bringing down this temperature feedback. See response above for details.

*Line 20 - add Biskaborn reference (https://www.nature.com/articles/s41467-018-08240-4)*

This is a useful reference, thank you. Added.

*Line 27/29 - Burke et al., 2017 (https://bg.copernicus.org/articles/14/3051/2017/bg-14-3051-2017.pdf)*

Added.

*Line 30 - JULES - Joint UK Land Environment Simulator*

Added.

*Line 31 - remove both 'can' s*

Thank you for catching the typo! Corrected.

*Line 35 - these more complex models still have missing/incorrectly parameterized processes and would definitely benefit from uncertainty quantification.*

We have added the fact that these models do also benefit from uncertainty quantification to the sentence as follows: "While high complexity models benefit from uncertainty quantification, they require large numbers of inputs and are computationally expensive, making it difficult to do uncertainty analysis directly with these models."

*Line 39 - remove 'models'*

Corrected.

*Line 45 - remove 'and'*

Corrected.

*Line 48 - The paragraph above suggests that these simple climate models do not have good spatio- temporal resolution but then this line suggests that Hector will be used to evaluate regional impacts.*

Unlike an ESM, Hector is not spatially explicit. However, Hector does have the ability to account for some spatial heterogeneity in land surface processes by dividing the global land surface into biomes, each with their own parameters, pools and fluxes. We have adjusted this sentence for clarity. It now reads: "Including a representation of permafrost in this model will allow for the consideration of permafrost in future such analyses with Hector, and, thanks to Hector's ability to represent separate biomes or regions, will be particularly important to evaluating the specific impacts of climate change in high latitudes."

*Line 71 - Equation 1 defines a land-to-atmosphere flux but Equation 3 suggests an atmosphere-to- land flux.*
*Line 71 - I think the last sign in Equation 1 should be + and the definition of FL should specifically state that NPP and Respiration act in opposite directions*
*Line 73 - FL is the difference between NPP (uptake) and respiration (loss).*

The land flux should be understood as positive into the land, and we have added language to clarify this. We have additionally corrected the language and sign in the $F_L$ equation and description to make it clear that NPP and RH act in opposite directions. We have kept the negative sign in Eqn 1 to be consistent with an understanding of this term being defined as positive into the land.

*Line 78/79 - what are the 1/4 and 1/50 factors for?*

These are the annual fractions of respiration carbon transferred to detritus and soil, respectively. We have replaced these with $f_{rd}$ and $f_{rs}$ and defined these terms in the paragraph following these

equations.

*Equation 4 has an air temperature change as a power and Equation 5 - has a running mean air temperature as a power? That looks a bit odd?*

Equation 4 is merely the canonical $Q_{10}$ response of biological processes (in this case, heterotrophic respiration from the land to the atmosphere) to temperature (see for example Davidson and Janssens 2006 - https://onlinelibrary.wiley.com/doi/abs/10.1111/j.1365-2486.2005.01065.x). Equation 5 is intended to provide a buffered response of soil temperature (and thus decomposition) to air temperature changes, and we agree that it's a highly empirical and only an approximation. We now note in the discussion that this should be an area of future focus and development for the Hector model.

*Line 85 - Where does the 200 years come from? Can this be justified further?*

The 200 year value for this smoothing is somewhat arbitrary. We have added this language to the sentence as follows: "Detritus respiration increases with region-specific air temperature change (T[i,t]) while soil respiration increases with the 200-year running mean of air temperature ($T_{200}$ [i,t]), a somewhat arbitrary choice of smoothing used in Hector as a proxy for soil temperatures." See also response above; we have added a note about this in the discussion.

*Line 99 - Equation 6/7 - DCperm is added to both Cperm and Cthawed? Surely it should be subtracted from one and added to the other?*

We have corrected Equation 6 to subtract this term.

*Line 99 - What is Fthawed-atm? Is the sign correct?*

We have corrected the sign and added the following language to explain this term: "$F_{thawed-atm}$ is the flux of carbon, in Pg C, from the thawed permafrost pool to the atmosphere, including both $CO_2$ and $CH_4$ emissions."

*Line 103 - shouldn't PHI be a volume fraction?*

Phi is a mass fraction of permafrost carbon, as Hector does not directly compute permafrost area or volume. We have added that language for clarity.

*Line 109 - Please give more details on Equation 9 and explore its validity.*

We have added the following language to the methods section: "The lognormal CDF was chosen for several reasons. Its curvature captures the "activation energy" of permafrost thaw with respect to temperature for low temperature change (left side of the curve), and, more importantly, the "diminishing returns" of permafrost thaw at higher temperatures because the

more accessible near-surface permafrost has already thawed by that point. Additionally, its parameters are readily interpretable in terms of the timing of 50% permafrost loss (exp(mu)) and the rate of permafrost loss around the 50% point relative to earlier/later in the process (sigma), which facilitates the use of this framework to emulate global permafrost dynamics in more complex models. Finally, it is naturally bounded between 0 and 1, which is appropriate as a model of the remaining permafrost fraction.

There are a variety of possible choices for this functional form and others can be explored in future model development efforts. Fortunately, the modular design and coding best practices of Hector make it simple to substitute this equation with alternatives."

We will also compare this relationship to that derived from CMIP6 data in our revised manuscript.

*Line 139 - please state earlier that the 308 Pg C soil carbon is 'non-permafrost' carbon*

We have clarified the land pools that are created when permafrost is added, specifying that soil is non-permafrost soil C at the beginning of the permafrost methods section and have specified non-permafrost soil when it is mentioned after this point.

*Table 1 caption - it says mu and sigma are from Koven et al, but in the Tabel it says they are from Kessler.*

We have clarified this language to read: "...estimated by optimizing the model against results from Koven *et al (*2013) while keeping within the upper and lower bounds from Kessler (2017)."

*Table 1 - no range for Cosoil or Cveg - have the authors checked if the model is sensitive to these?*

We have not checked the model's sensitivity to these parameters as we have focused our sensitivity analysis on parameters we have added to the model, such as the methane fraction and temperature controls on permafrost thaw. In our revised manuscript we will use CMIP6 outputs to derive ranges for these numbers.

*Table 1/Line 144 - what is the wf used for? I can't see it in any equation? How does this compare to the value in Chadburn et al. 2017 (https://www.nature.com/articles/nclimate3262)*

We have added an equation to define this term better (new Equation 6). This value (2.0) is lower than those used in Chadburn *et al.* (2017) though higher than that used in some earlier models such as MacDougall *et al.* (2012).

*Table 1 - f_RH_CH4 does not match the name in equation 11/12*

We have corrected this name to read f$_{CH4}$ throughout the text.

*Line 150 - cite Burke et al. 2020 (https://tc.copernicus.org/articles/14/3155/2020/tc-14-3155-2020.pdf) which shows CMIP6 and CMIP5 are very similar.*

In our revised manuscript we will update to use CMIP6 data instead of CMIP5.

*Table 2 - how does this compare to Burke et al. 2020?*

We will update Table 2 to use our own analysis of CMIP6 data in our revised manuscript.

*Line 155 - why do we expect mu/sigma to fall within the range of Kessler 2017?*

To clarify, we used Kessler 2017 as a *range* for these two parameters, as the reviewer notes, and estimated the mu and sigma within this range that produced a best-fit result relative to CMIP5 (CMIP6 in the revision). We could of course let the parameters exceed this range, but feel that this nonetheless provided a useful *a priori* guide for this exploratory work.

*Line 159 - this 70 % was not included in the uncertainty range. Any reason?*

We have updated our default value of the static fraction to 72% (Schädel *et al.* 2014) and re-run our analysis. Our range is now defined around this baseline based on the Schädel paper.

*Line 161 - uncertainty ranges in Table 1 do not reach the 4.3 % suggested in the text. Why not?*

Thank you for catching this. We have updated the table and added the Schädel citation.

*Line 178 - check name f_RH_CH4*

Corrected to be consistent with previous equations.

*Line 179 - how were the parameters sampled from prior distribution? Latin Hypercube?*

They were uniformly sampled. We have added language to specify this.

*Line 181-184 - please give more details on what these parameters mean and what the approach of LeBauer is.*

We have added the following text to this section: "Briefly, the coefficient of variation describes the uncertainty in the parameter (calculated as the parameter variance divided by the mean), the elasticity describes the sensitivity of the model to a relative change in the parameter, and the partial variance synthesizes these two metrics to describe the relative contribution of uncertainty in a parameter to the total predictive uncertainty in the model output (i.e., the parameters that have the highest partial variance are those that are highly uncertain and to which the model is

highly sensitive; parameters that are highly uncertain but to which the model is relatively uncertain, and conversely, parameters to which a model is highly sensitive but whose values are known precisely, would both have low partial variance)."

And we elaborate on the approach of LeBauer as follows: "We generally follow the approach of Lebauer *et al.* (2013) which samples from parameter distributions to generate an ensemble of model runs that approximate the posterior distribution of model output that can be used in the sensitivity analysis. Their sensitivity analysis is based on univariate perturbations of each parameter of interest, and the relationship between each parameter and model output is approximated by a natural cubic spline. Their model sensitivity is then based on the derivative of the spline at the parameter median."

*Line 188 - these 300-400 Pg C are not yet decomposed so comprise the Cthawed pool?I*

These values include carbon that has been decomposed as well as that which is still in the thawed pool. We have clarified this. The sentence now reads: "In RCP 4.5, 6.0, and 8.5, permafrost losses, including both thawed permafrost and permafrost carbon that has been decomposed and emitted to the atmosphere, reached 300-400 Pg C by 2100 and mostly leveled off after this point."

*Line 195 - is there any evidence that the ~3 Pg C /year has been found in other models. This is quite high.*

We agree this is high compared to, for example, Burke *et al.* (2017). Since adjusting the default static fraction of thawed permafrost carbon in the model to 0.72, we find a substantially lower flux, closer to 1.5 Pg C/year.

*Figure 3 - Again I think the feedback temperature is generally quite high compared with other simulations. It would be good to see the spread introduced by including the parameter uncertainties.*

We have adjusted the model to use a higher static fraction of thawed permafrost, substantially reducing this temperature effect, and have also added a figure (see response above) to the sensitivity analysis section showing the sensitivity of temperature to the various permafrost controls in the model.

*Figure 3 - Please look up the standard colours for the RCP scenarios and use them. It is a little confusing to have RCP2.6 as red.*

We appreciate the tip about standard colors and have adjusted this figure accordingly.

*Line 221 - Quite a lot of this carbon remains in the atmosphere (expect ~50 %, ~25% to land/~25% to ocean). Is this a function of the model structure?*

We agree that this fraction is high compared to estimates of, e.g., the fraction of anthropogenic emissions that are estimated to remain in the atmosphere (Knorr 2009). Interestingly, however, more recent estimates of the airborne fraction sometimes estimate considerably higher values—see for example Yin et al. (2020). https://www.nature.com/articles/s41467-020-15852-2

Regardless, this potential discrepancy is interesting and will help Hector developers prioritize future efforts and model evaluation exercises. A current focus of these efforts is adding a carbon-tracking feature to the model, which will allow researchers to more clearly diagnose and evaluate the model's carbon flows in the future.

*Figure 5 looks interesting but is not immediately clear. Please include equation symbols in the names. I am not familiar with all of these statistics so a way to highlight the interesting ones and link them to the text would be great.*

We have simplified this figure to only include the coefficient of variation, elasticity, and partial variance in order to more closely follow the revised description of these statistics given in the methods section (see response above) and make it easier to follow. We have also included some of the description of these statistics from the revised methods section into the figure caption as follows: "The coefficient of variation describes the uncertainty in the parameter (parameter variance divided by the mean), the elasticity describes the sensitivity of the model to a relative change in the parameter, and the partial variance synthesizes these two metrics to describe the relative contribution of uncertainty in a parameter to the total predictive uncertainty in the model output."

[Figure]

*Line 235 – the 30-45% is 'partial variance'?*

Yes. We have clarified this sentence to reflect that.

*Line 266 – Walter-Anthony, 2018*

Added.

*Line 275 – this should be encompassed by the parameter uncertainty.*

This is an interesting point—but we don't believe it would be included by current parameter uncertainty, because the solution would be to add a completely new parameter controlling the decomposition rate of thawed permafrost (versus non-permafrost soil). In any event, we have removed this paragraph in the revised manuscript.

*Line 275 – 280 – this relates to the abrupt thaw processes discussed above and the two discussed together.*

We have moved this paragraph up in section 4.1 and have connected it better to abrupt thaw.

*Line 304 – Chadburn et al assumed the soil and air temperatures were in equilibrium in their analysis.*

We appreciate this point of clarification and have added the following language to that sentence in the discussion: "though this analysis used equilibrium temperatures and does not give us information about the potential for these insulation effects to play a role in mitigating transient thaw."

*How well does the 200 year temperature term represent the thermal inertia of the permafrost?*

It actually is not intended to represent the thermal inertia of permafrost, as this value is only used for estimating respiration from thawed soil. We regret that this was not clear in the paper and have added some clarifying language. See response above for additional comments on this point.

*Section 4.1 probably need to mention nutrient limitation.*

We appreciate this suggestion and have added a mention of nutrient limitation as follows: "Our results exclude any potential changes in plant productivity as a result of permafrost thaw, including any due to changes in nutrient availability, as the sign of these effects remains highly uncertain."

*Table 4 – add the results of the temperature effect here. I am not sure why some comparisons are in the table and some are in the text.*

We have added an additional line to Table 4, comparing against MacDougal *et al.* (2012) and have simplified the text to be consistent with the table.

*Line 334 – please define/reference GCAM.*

Added.

**CEC1 comments**

*- The code of the model must be stored in a permanent archive—for example, Zenodo. Github is not a repository acceptable for purposes of long-term storage.*

We will add the code to a Zenodo archive to be cited in the revised manuscript.

*- You must include the version of the model that you use in the title of the manuscript.*

We will revise the title to include the model version number.

---

## Author Response (AR1)

**A Permafrost Implementation in the Simple Carbon-Climate Model Hector v2.3pf**

Dawn L. Woodard, Alexey N. Shiklomanov, Ben Kravitz, Corinne Hartin, and Ben Bond-Lamberty

Dear editors and reviewers:

We appreciate the extensive comments we have received on this manuscript, and have completed a top-to-bottom revision of our manuscript in an effort to thoroughly address the reviewers' concerns and suggestions. We respond individually to each reviewer comment below, describing these updates in more detail, and summarize the major changes here for clarity.

Key model changes included in this revision:
- Corrected our handling of the static fraction of thawed permafrost carbon in Hector (see Equation 12, Section 2.1.1)
- Updated the static fraction default parameter to 74% from 40% based on a more recent reference
- Retuned our permafrost thaw parameters to correspond to the static fraction update and revised approach to estimating their ranges
- Adjustments to other parameters: initial permafrost, vegetation, and detritus carbon in the permafrost region
- These parameter and structural changes have slowed the permafrost carbon emissions and substantially reduced estimates of permafrost-driven temperature change and effects on atmospheric carbon dioxide in the near term.

Key text and figure updates:
- Added two additional parameters to our sensitivity analysis: the warming factor and the total non-permafrost carbon in the permafrost region and adjusted the ranges used in the sensitivity analysis for the methane fraction to include results from additional studies
- Added Figure 6 as part of our sensitivity analysis to demonstrate the concrete impacts on temperature of varying each parameter across its potential range from the literature
- Added further references and comparisons in Table 4 to better place our results in the context of existing studies.
- Expanded Figure 2 to include results from previous studies as well as CMIP6 model results to support the relationship of permafrost thaw versus temperature used in Hector.
* * *
**RC1 comments**

*This study projects permafrost thaw and associated GHG emissions using a new representation of permafrost, which was integrated to the global carbon-climate model Hector. The authors use air temperature projections to quantify permafrost thaw. They acknowledge the limitations that*

*come with the use of a simple model such as theirs, and do not over-interpret their model outputs.*

*I enjoyed reading this manuscript and believe it will be of broad interest to the scientific community as well as informative to IAMs. The Discussion section does a great job at documenting the model limitations (as a field ecologist, I appreciate that); I also think the section that compared this study's model outputs with that of other models was beneficial to the reader.*

We appreciate the reviewer's interest and overall positive assessment.

*I'd be interested to see a few more details pertaining to: (1) How is the "static" (non-labile) C fraction in the permafrost determined? (2) Why is the CH4 emission fraction from thawed permafrost set to 2.3% (please add references or some kind of explanation), and how much effect would a lower or higher fraction have on GHG (provide graphs)? (3) How sensitive is the model to the permafrost pool size (provide graphs of projected GHG under different pool sizes)?*

1) We have added the following sentence starting on line 141 to better explain where the non-labile fraction comes from:

"While in reality turnover times of soil organic carbon fall anywhere along the range from a few days to thousands of years (Schädel et al., 2014), we group soil decomposition broadly into labile and non-labile pools, where carbon in the non-labile (static) pool decomposes on the order of up to thousands of years and is assumed to be inert for the purpose of this analysis."

2) We have added references to support this parameter choice and expanded the range in Table 1 based on additional results. We have additionally added a new figure (Figure 6) to visualize the sensitivity of the model in terms of the magnitude of the impact on temperature to the potential ranges of all our key parameters from the sensitivity results shown in Figure 5.
3) We have added Figure 6 to demonstrate this sensitivity more concretely, which we focus on temperature, rather than methane and carbon dioxide in order to capture the effects on both.

*Minor comments:*

*If possible, add Hugelius et al 2020 (PNAS; https://www.pnas.org/content/117/34/20438) to Table 4, and discuss their findings in light of yours.*

As our results do not separate out peatlands, it is not straightforward to directly compare our results to this paper in Table 4, but we have added the following starting on line 365 in our discussion addressing the implications of Hugelius *et al.* 2020 for our results.

"Abrupt thaw is also a key process for permafrost in peatland soils, and a recent analysis estimates an additional 40 Pg of permafrost carbon stored in peat than had been found

previously (Hugelius et al., 2020). Based on our sensitivity analysis, increasing the initial permafrost by this amount might translate to around a 0.02°C increase in overall temperature change by 2100."

*l-39: "models" is there twice*

Thanks for the catch! Corrected on line 40.

*l-181: fix the typo in the word "parameter"*

Corrected on line 242.

**RC2 comments**

*This paper discusses the implementation of a simple permafrost carbon module within the Hector simple climate model. The authors talk about uncertainties but the results have very few details of their impact.*

We appreciate the reviewer's careful examination of our paper and thoughtful comments here and below.

We have added a figure (Figure 6) showing the impact of parameter uncertainties on global mean temperature; this output variable is integrative, capturing the effects on other key outcomes as well (permafrost thaw, methane and carbon dioxide emissions). We have also expanded further on uncertainties in the discussion (described further in various responses below).

*In particular, the model results have a high permafrost carbon feedback temperature compared to other results in the literature. I think this is fine if the uncertainties are made more high profile throughout the whole document particularly plumes in Figure 3 but also maybe table 3. Q quick thought - do the authors think this high feedback temperature is caused by a relatively large methane contribution?*

We agree that the temperature response in the original version of the manuscript was high compared to many existing estimates. In our revised manuscript we updated the non-labile fraction to a higher value of 0.74 based on the more recent and comprehensive Shädel *et al.* (2014), compared to the previous value of 0.4, though we still include 0.4 as a lower bound in our sensitivity analysis. We also revise the tuning of our permafrost thaw parameters based on this and other parameter updates. These changes end up bringing our temperature response down to ~0.2-0.25 °C by 2100 across all scenarios. We have also checked the model parameterization of permafrost thaw versus temperature against CMIP6 data to add further validation of this relationship. We note that one of Hector's strengths is in its ability to calibrate to different assumptions and explore their consequences, so that our default configuration should only be seen as a starting point for looking at the effects of permafrost in this model.

We additionally have added acknowledgement of the comparison between our temperature estimates and those from previous results in our results section on line 300. While our estimate is no longer higher than previous studies, we do recognize the significant contribution of methane to temperature in Hector and have added the following sentence to line 433 comparing the estimated methane-driven warming to that in previous studies.

"The $CH_4$ contribution to permafrost-driven temperature change estimated by Hector was between 24 and 29%, somewhat higher than the 16% given in Schaefer et al. (2014), but just under the 30-50% range given by the expert assessment in Schuur et al. (2013)."

*I think more details and discussion are required about the physical permafrost change parametrisation. This is a key addition to the model and not explored in much detail. I see it is discussed more in Figure 2 but I think it needs to be compared to something other than Kessler.*

This is a good point, and we have expanded Figure 2 substantially to include other projections of high latitude temperature change versus permafrost thaw fraction.

***Why is a volume fraction not used when considering physical permafrost change?*** *This is much more relate-able to carbon amount.*

Because of Hector's structure and global spatial scale, the permafrost component is based only on the movement of carbon by mass, with no explicit representation of depth or area, so this is most accurately thought of as a permafrost carbon mass fraction. We have specified this in the methods section on lines 100 and 115.

*Please check the signs/definition of fluxes in the equations and their text.*

We have carefully gone through the paper and checked signs and added clarity to the definitions of our fluxes to ensure consistency.

*It would be good to have units included for all variables. Also check all acronyms are defined.*

We have double-checked the paper to add units where missing and definitions of acronyms before use.

*In general the paper contains most of the required information but sometimes later than I would like it.*

This is helpful feedback. We have attempted to clarify our language and ensure the relevant information is given early enough in the paper.

*Minor comments:*
*Line 3 - permafrost C feedback is hardly ever estimated using ESMs currently.*

We have edited the sentence on line 3 to reflect this clarification.

*Line 6 - ?? ESM permafrost estimate*

As we tuned to several different values, we have left the details of these estimates for later in the manuscript in favor of simplicity within the abstract.

*Line 10 - 0.5 degree feedback temperature is relatively high.*

It is, and while diverging from existing estimates is not necessarily cause for concern, as described above, the adjustment we have made to our static fraction have ended up bringing down this temperature feedback.

*Line 20 - add Biskaborn reference (https://www.nature.com/articles/s41467-018-08240-4)*

This is a useful reference, thank you. Added on line 21.

*Line 27/29 - Burke et al., 2017 (https://bg.copernicus.org/articles/14/3051/2017/bg-14-3051-2017.pdf)*

Added on line 30.

*Line 30 - JULES - Joint UK Land Environment Simulator*

Added on line 32.

*Line 31 - remove both 'can' s*

Thank you for catching the typo! Corrected on lines 32/33.

*Line 35 - these more complex models still have missing/incorrectly parameterized processes and would definitely benefit from uncertainty quantification.*

We have added the fact that these models do also benefit from uncertainty quantification to the sentence on line 35.

*Line 39 - remove 'models'*

Corrected on line 40.

*Line 45 - remove 'and'*

Corrected on line 48.

*Line 48 - The paragraph above suggests that these simple climate models do not have good spatio- temporal resolution but then this line suggests that Hector will be used to evaluate regional impacts.*

Unlike an ESM, Hector is not spatially explicit. However, Hector does have the ability to account for some spatial heterogeneity in land surface processes by dividing the global land surface into biomes, each with their own parameters, pools and fluxes. We have adjusted this sentence for clarity on lines 50/51.

The sentence now reads: "Including a representation of permafrost in this model will allow for the consideration of permafrost in future such analyses with Hector, and, thanks to Hector's ability to represent separate biomes or regions, will be particularly important for evaluating the specific impacts of climate change in high latitudes."

*Line 71 - Equation 1 defines a land-to-atmosphere flux but Equation 3 suggests an atmosphere-to- land flux.*
*Line 71 - I think the last sign in Equation 1 should be + and the definition of FL should specifically state that NPP and Respiration act in opposite directions*
*Line 73 - FL is the difference between NPP (uptake) and respiration (loss).*

The land flux should be understood as positive into the land, and we have added language to clarify this on line 76. We have additionally corrected the language and sign in the $F_L$ equation and description to make it clear that NPP and RH act in opposite directions. We have kept the negative sign in Eqn 1 to be consistent with an understanding of this term being defined as positive into the land.

*Line 78/79 - what are the 1/4 and 1/50 factors for?*

These are the annual fractions of respiration carbon transferred to detritus and soil, respectively. We have replaced these with $f_{rd}$ and $f_{rs}$ and defined these terms in the paragraph following these equations starting on line 86.

*Equation 4 has an air temperature change as a power and Equation 5 - has a running mean air temperature as a power? That looks a bit odd?*

Equation 4 is merely the canonical $Q_{10}$ response of biological processes (in this case, heterotrophic respiration from the land to the atmosphere) to temperature (see for example Davidson and Janssens 2006 - https://onlinelibrary.wiley.com/doi/abs/10.1111/j.1365-2486.2005.01065.x). Equation 5 is intended to provide a buffered response of soil temperature (and thus decomposition) to air temperature changes, and we agree that it's a highly empirical and only an approximation. We now note in the discussion on lines 404-407 that this should be an area of future focus and development for the Hector model.

*Line 85 - Where does the 200 years come from? Can this be justified further?*

The 200 year value for this smoothing is somewhat arbitrary. We have added this language to the sentence on lines 88/89 as follows: "Detritus respiration increases with group-specific air temperature change ($T[i,t]$), while soil respiration increases with the 200-year running mean of air temperature ($T_{200}[i,t]$), a somewhat arbitrary choice of smoothing used in Hector as a proxy for soil temperatures." See also response above; we have added a note about this in the discussion.

*Line 99 - Equation 6/7 - DCperm is added to both Cperm and Cthawed? Surely it should be subtracted from one and added to the other?*

We have corrected Equation 7 on line 109 to subtract this term.

*Line 99 - What is Fthawed-atm? Is the sign correct?*

We have corrected the sign and added some language (below) on lines 111/112 to explain this term, as well as including an equation for it (equation 16) on line 166 once the other relevant fluxes have been explained.

"$F_{thawed-atm}$ is the flux of carbon, in Pg C, from the thawed permafrost pool to the atmosphere, including both $CO_2$ and $CH_4$ emissions (see section 2.1.1)."

*Line 103 - shouldn't PHI be a volume fraction?*

Phi is a mass fraction of permafrost carbon, as Hector does not directly compute permafrost area or volume. We have added that language to line 115 for clarity and have also changed the parameter name to $f_{frozen}$ to be consistent with language used later in the manuscript.

*Line 109 - Please give more details on Equation 9 and explore its validity.*

We appreciate the feedback that this was insufficiently described. We have expanded Figure 2 substantially to validate this relationship against the literature and CMIP6 models and have added the following language to the methods section starting on line 130:

 "The lognormal CDF was chosen for several reasons. Its curvature captures the "activation energy" of permafrost thaw with respect to temperature for low temperature change (left side of the curve), and, more importantly, the "diminishing returns" of permafrost thaw at higher temperatures because the more accessible near-surface permafrost has already thawed by that point. Additionally, its parameters are readily interpretable in terms of the timing of 50% permafrost loss (exp(mu)) and the rate of permafrost loss around the 50% point relative to earlier/later in the process (sigma), which facilitates the use of this framework to emulate global permafrost dynamics in more complex models. Finally, it is naturally bounded between 0 and 1, which is appropriate as a model of the remaining permafrost fraction.

There are a variety of possible choices for this functional form and others can be explored in future model development efforts. Fortunately, the modular design and coding best practices of Hector make it simple to substitute alternatives for this equation."

*Line 139 - please state earlier that the 308 Pg C soil carbon is 'non-permafrost' carbon*

We have clarified the land pools that are created when permafrost is added, specifying that soil is non-permafrost soil C at the beginning of the permafrost methods section on line 99 and have specified non-permafrost soil when it is mentioned after this point.

*Table 1 caption - it says mu and sigma are from Koven et al, but in the Tabel it says they are from Kessler.*

We have clarified this language in the table caption to read: "...estimated by optimizing the model against results from Koven *et al (*2013) while keeping within the upper and lower bounds from Kessler (2017)."

*Table 1 - no range for Cosoil or Cveg - have the authors checked if the model is sensitive to these?*

In the revised manuscript we use CMIP6 outputs to derive ranges for these numbers, now shown in Table 1, and have added the total non-permafrost carbon in the permafrost region to our sensitivity analysis (spoiler: it turns out that no, the model is not particularly sensitive to changes in these values).

*Table 1/Line 144 - what is the wf used for? I can't see it in any equation? How does this compare to the value in Chadburn et al. 2017 (https://www.nature.com/articles/nclimate3262)*

We have added an equation to define this term better (new Equation 6, line 94). This value (2.0) is lower than those used in Chadburn *et al.* (2017) though higher than that used in some earlier models such as MacDougall *et al.* (2012).

*Table 1 - f_RH_CH4 does not match the name in equation 11/12*

We have corrected this name to read $f_{CH4}$ throughout the text.

*Line 150 - cite Burke et al. 2020 (https://tc.copernicus.org/articles/14/3155/2020/tc-14-3155-2020.pdf) which shows CMIP6 and CMIP5 are very similar.*

We have added on lines 200-202 that these fractions are substantially similar between CMIP5 and CMIP6 based on our own analysis and as also found in Burke *et al*. (2020).

*Table 2 - how does this compare to Burke et al. 2020?*

We chose to use our own analysis of CMIP6 data instead of attempting to estimate these fractions from the Burke paper. The differences were not substantial enough to justify tuning the model to them instead, which we note in the text on lines 200-202, and thus we have left the CMIP5 results in Table 2 as the actual results we tuned against, having the advantage of being already published.

*Line 155 - why do we expect mu/sigma to fall within the range of Kessler 2017?*

To clarify, we used Kessler 2017 as a *range* for these two parameters, as the reviewer notes, and estimated the mu and sigma within this range that produced a best-fit result relative to CMIP data. We could of course let the parameters exceed this range, but feel that this nonetheless provided a useful *a priori* guide for this exploratory work.

*Line 159 - this 70 % was not included in the uncertainty range. Any reason?*

We have updated our default value of the static fraction to 74% (Schädel *et al.* 2014) and re-run our analysis. Our range for the uncertainty analysis is now defined around this baseline based on the Schädel paper, but we do still include the previous value of 40% as a lower bound.

*Line 161 - uncertainty ranges in Table 1 do not reach the 4.3 % suggested in the text. Why not?*

Thank you for catching this. This was an erroneous calculation, and we do now include a different result from that paper as our lower bound on the methane fraction and a somewhat higher upper bound from a different paper, which should now be consistent across the text and table.

*Line 178 - check name f_RH_CH4*

Corrected on line 235 to be consistent with previous equations.

*Line 179 - how were the parameters sampled from prior distribution? Latin Hypercube?*

They were randomly sampled. We have added language to specify this on line 241.

*Line 181-184 - please give more details on what these parameters mean and what the approach of LeBauer is.*

We have added the following text to this section starting on line 245.

 "Briefly, the coefficient of variation describes the uncertainty in the parameter (calculated as the parameter variance divided by the mean), the elasticity describes the sensitivity of the model to a relative change in the parameter, and the partial variance synthesizes these two metrics to

describe the relative contribution of uncertainty in a parameter to the total predictive uncertainty in the model output (i.e., the parameters that have the highest partial variance are those that are highly uncertain and to which the model is highly sensitive; parameters that are highly uncertain but to which the model is relatively uncertain, and conversely, parameters to which a model is highly sensitive but whose values are known precisely, would both have low partial variance)."

And we elaborate on the approach of LeBauer as follows starting on line 252:

 "We generally follow the approach of Lebauer *et al.* (2013) which samples from parameter distributions to generate an ensemble of model runs that approximate the posterior distribution of model output that can be used in the sensitivity analysis. Their sensitivity analysis is based on univariate perturbations of each parameter of interest, and the relationship between each parameter and model output is approximated by a natural cubic spline. Their model sensitivity is then based on the derivative of the spline at the parameter median."

*Line 188 - these 300-400 Pg C are not yet decomposed so comprise the Cthawed pool?I*

These values include carbon that has been decomposed as well as that which is still in the thawed pool. We have clarified this. The sentence now reads: "In RCP 4.5, 6.0, and 8.5, permafrost losses, including both thawed permafrost and permafrost carbon that has been decomposed and emitted to the atmosphere, reached 350-450 Pg C by 2100, with the rate of thaw fastest over the 21st century and slowing thereafter."

*Line 195 - is there any evidence that the ~3 Pg C /year has been found in other models. This is quite high.*

We agree this is high compared to, for example, Burke *et al.* (2017). Since adjusting the default static fraction of thawed permafrost carbon in the model to 0.74, we find a lower flux, closer to 2 Pg C/year, which is still higher than Burke's results, but not by as much.

*Figure 3 - Again I think the feedback temperature is generally quite high compared with other simulations. It would be good to see the spread introduced by including the parameter uncertainties.*

We have adjusted the model to use a higher static fraction of thawed permafrost, substantially reducing this temperature effect, and have also added a figure (see response above) to the sensitivity analysis section showing the sensitivity of temperature to the various permafrost controls in the model.

*Figure 3 - Please look up the standard colours for the RCP scenarios and use them. It is a little confusing to have RCP2.6 as red.*

We appreciate the tip about standard colors and have adjusted this figure accordingly.

*Line 221 - Quite a lot of this carbon remains in the atmosphere (expect ~50 %, ~25% to land/~25% to ocean). Is this a function of the model structure?*

We agree that this fraction is high compared to estimates of, e.g., the fraction of anthropogenic emissions that are estimated to remain in the atmosphere (Knorr 2009). Interestingly, however, more recent estimates of the airborne fraction sometimes estimate considerably higher values—see for example Yin et al. (2020). https://www.nature.com/articles/s41467-020-15852-2

Regardless, this potential discrepancy is interesting and will help Hector developers prioritize future efforts and model evaluation exercises. A current focus of these efforts is adding a carbon-tracking feature to the model, which will allow researchers to more clearly diagnose and evaluate the model's carbon flows in the future.

*Figure 5 looks interesting but is not immediately clear. Please include equation symbols in the names. I am not familiar with all of these statistics so a way to highlight the interesting ones and link them to the text would be great.*

We have simplified this figure to only include the coefficient of variation, elasticity, and partial variance in order to more closely follow the revised description of these statistics given in the methods section (see response above) and make it easier to follow. We have also included some of the description of these statistics from the revised methods section into the figure caption:

 "The coefficient of variation describes the uncertainty in the parameter (parameter variance divided by the mean), the elasticity describes the sensitivity of the model to a relative change in the parameter, and the partial variance synthesizes these two metrics to describe the relative contribution of uncertainty in a parameter to the total predictive uncertainty in the model output."

*Line 235 – the 30-45% is 'partial variance'?*

Yes. We have clarified this sentence on line 325 to reflect that.

*Line 266 – Walter-Anthony, 2018*

Added on line 365.

*Line 275 – this should be encompassed by the parameter uncertainty.*

This is an interesting point—but we don't believe it would be included by current parameter uncertainty, because the solution would be to add a completely new parameter controlling the decomposition rate of thawed permafrost (versus non-permafrost soil). In any event, we have removed this paragraph in the revised manuscript.

*Line 275 – 280 – this relates to the abrupt thaw processes discussed above and the two discussed together.*

We have moved this paragraph up in section 4.1 starting on line 371 and have connected it better to abrupt thaw.

*Line 304 – Chadburn et al assumed the soil and air temperatures were in equilibrium in their analysis.*

We appreciate this point of clarification and have added clarity on that point to that sentence in the discussion on line 413.

*How well does the 200 year temperature term represent the thermal inertia of the permafrost?*

It actually is not intended to represent the thermal inertia of permafrost, as this value is only used for estimating respiration from thawed soil, while permafrost thaw is estimated based on a relationship with high latitude air temperatures. We regret that this was not clear in the paper and have added some clarifying language on line 88. See response above for additional comments on this point.

*Section 4.1 probably need to mention nutrient limitation.*

We appreciate this suggestion and have added a mention of nutrient limitation on line 404.

*Table 4 – add the results of the temperature effect here. I am not sure why some comparisons are in the table and some are in the text.*

We have updated Table 4 to include several additional results, including comparing against MacDougal *et al.* (2012), and have simplified the text (lines 438-443) to be consistent with the table.

*Line 334 – please define/reference GCAM.*

Added to line 451.

**CEC1 comments**

*- The code of the model must be stored in a permanent archive—for example, Zenodo. Github is not a repository acceptable for purposes of long-term storage.*

We have added the model code to a Zenodo archive now cited in the code availability section.

*- You must include the version of the model that you use in the title of the manuscript.*

We have revised the title to include the model version number.